**Data Availability Statement:** All data, codes, and files necessary for replicating this study are

# Particle swarm optimization solution for roll-off control in radiofrequency ablation of liver tumors: Optimal search for PID controller tuning

**Rafael Mendes Faria**[1,2☯*], **Suélia de Siqueira Rodrigues Fleury Rosa**[1,3☯], **Gustavo Adolfo Marcelino de Almeida Nunes**[1☯], **Klériston Silva Santos**[1,2☯], **Rafael Pissinati de Souza**[1,4☯], **Angie Daniela Ibarra Benavides**[1☯], **Angélica Kathariny de Oliveira Alves**[1☯], **Ana Karoline Almeida da Silva**[1☯], **Mario Fabrício Rosa**[3☯], **Antônio Aureliano de Anicêsio Cardoso**[3☯], **Sylvia de Sousa Faria**[5☯], **Enrique Berjano**[5☯], **Adson Ferreira da Rocha**[6☯], **Ícaro dos Santos**[7☯], **Ana González-Suárez**[8☯]

1 Department of Mechanical Engineering, University of Brasilia, Brasilia, Distrito Federal, Brazil, 2 Department of Electrical Engineering, Federal Institute of Education, Science and Technology of Triângulo Mineiro, Paracatu, Minas Gerais, Brazil, 3 Department of Biomedical Engineering, Faculty of Gama, University of Brasilia, Brasilia, Distrito Federal, Brazil, 4 Department of Electrical Engineering, Federal Institute of Education, Science and Technology of Rondônia, Porto Velho, Rondônia, Brazil, 5 Department of Electronic Engineering, Universitat Politècnica de València, Valencia, Spain, 6 Department of Electrical Engineering, University of Brasilia, Brasilia, Distrito Federal, Brazil, 7 Department of Electrical Engineering and Computer Science, Milwaukee School of Engineering, Milwaukee, Wisconsin, United States of America, 8 Translational Medical Device Lab, School of Medicine, University of Galway, Galway, Ireland

☯ These authors contributed equally to this work.
* rafaelmendes@iftm.edu.br

## Abstract

The study investigates the efficacy of a bioinspired Particle Swarm Optimization (PSO) approach for PID controller tuning in Radiofrequency Ablation (RFA) for liver tumors. Ex-vivo experiments were conducted, yielding a $9^{th}$ order continuous-time transfer function. PSO was applied to optimize PID parameters, achieving outstanding simulation results: 0.605% overshoot, 0.314 seconds rise time, and 2.87 seconds settling time for a unit step input. Statistical analysis of 19 simulations revealed PID gains: $Kp$ (mean: 5.86, variance: 4.22, standard deviation: 2.05), $Ki$ (mean: 9.89, variance: 0.048, standard deviation: 0.22), $Kd$ (mean: 0.57, variance: 0.021, standard deviation: 0.14) and ANOVA analysis for the 19 experiments yielded a p-value $\ll 0.05$. The bioinspired PSO-based PID controller demonstrated remarkable potential in mitigating roll-off effects during RFA, reducing the risk of incomplete tumor ablation. These findings have significant implications for improving clinical outcomes in hepatocellular carcinoma management, including reduced recurrence rates and minimized collateral damage. The PSO-based PID tuning strategy offers a practical solution to enhance RFA effectiveness, contributing to the advancement of radiofrequency ablation techniques.

included in the Compressed/ZIP Supporting information file.

**Funding:** This study was supported by the University of Brasília (UnB) and the Fundação de Empreendimentos Científicos e Tecnológicos (FINATEC), project number 7426. We also thank the support of Federal Deputy Erika Kokay.

**Competing interests:** The authors have declared that no competing interests exist.

## Introduction

Cancer stands as a significant global concern, ranking among the top four leading causes of premature mortality before the age of 70 in many countries [1]. Worldwide, an estimated 18 million new cancer cases occur annually, with a higher incidence in the male population, accounting for approximately 53% of the total [1, 2]. Among various cancer types, liver tumors are the second most lethal, with an incidence rate of approximately 6.3% among newly diagnosed cases [3, 4].

HCC, a form of liver cancer significantly influenced by the tumor stage, exhibits a five-year survival rate exceeding 70% for early-stage cases, while the average survival varies from 1 to 1.5 years for advanced and symptomatic cases treated with systemic therapies [3]. Accurate diagnosis distinguishing HCC from other liver diseases with similar clinical and radiological features is crucial for effective treatment planning [5].

The treatment of HCC presents challenges due to tumor resistance and recurrence [6]. Early stages are addressed through surgery, while advanced stages can be treated with chemotherapy, immunotherapy, oncolytic viruses, and nanotechnology to enhance effectiveness and reduce side effects [7, 8]. However, these methods face substantial challenges that limit their effectiveness. Hepatic resection and liver transplantation are effective in early stages but are not options for most patients. Traditional chemotherapy is limited by drug resistance, and radiotherapy is constrained due to the liver's sensitivity to radiation. Immunotherapy and oncolytic viruses show promise but do not benefit all patients [8–11]. Additionally, RFA therapy, although effective in some cases, has its own limitations, including restrictions on the size and location of treatable tumors. Identifying risk factors and trends is crucial for developing effective strategies for prevention and intervention in the fight against liver cancer [11].

The importance of diversified techniques in the treatment of HCC is essential for enhanced treatment efficacy and safety. Thermal ablation is a minimally invasive medical procedure that utilizes microwave, laser, radiofrequency (RF), or ultrasound energy to perform tissue ablation [5, 12]. In RFA, the procedure entails the use of an ablation electrode that is navigated to the target region utilizing imaging techniques such as computed tomography or ultrasonography. Subsequently, the ablation is performed via the delivering of RF energy in the target region [13–17]. The Joule effect is the mechanism by which electromagnetic energy is converted into thermal energy during the RF ablation procedure [18]. As such, the intense heat generated through ionic agitation of molecules results in cellular demise and eventual necrosis of tumor cells. [19].

However, ablation procedures are subject to certain limitations, particularly in achieving precise and complete coagulation volumes that fully encompass larger tumors [20, 21]. The search for strategies aimed at enhancing the ablation volume ensued due to the occurrence of a limited coagulation zone in the target region [22]. The infusion of physiological solutions is currently utilized with the objective of augmenting tissue conductivity and promoting uniform energy propagation [23].

Research conducted by Jiang *et al.* [24] reveals that in the context of liver RFA, the adoption of internally cooled electrodes in conjunction with the injection of 10% hydrochloric acid (HCl) markedly augments the ablation zone volumes, thereby enabling the treatment of larger tumors. This increase in efficacy can be attributed to improved thermal conductivity, amplified electrical conductivity, and reduced impedance. Notwithstanding, other studies involving the utilization of infused saline solutions report a delayed onset of roll-off, which is postulated to be caused by the tissue hydration provided by the solution, leading to a reduction in dehydration and localized vaporization [13].

The liver comprises hepatic arteries, portal veins, and hepatic veins, each characterized by distinct diameters, flow rates, and spatial arrangements [15, 20, 21]. Although effective, RFA is

hampered by certain technological constraints, resulting in complications such as local recurrences, deficient ablation zone control in the vicinity of the targeted field, risk of harm to non-tumoral tissues, and potential harm to vascular and biliary structures [14, 15, 18, 25–27].

As a consequence of ablative effects, a study conducted by [28] presents a significant reduction (44.4±14.6%) in tumor volume following RFA in HCC patients. In this context, ex vivo assays by [29] demonstrated a shrinkage in the coagulation zone in hepatic (15–31%) and pulmonary tissues. On the other hand, [30] conducted in silico assays applied to breast cancer and showed that with an increase in electrical potential, there was a significant growth in temperature and the value of the Arrhenius integral, which estimates the degree of tissue destruction. In this study, [30] also demonstrated that the optimal voltage level for the ablative procedure was between 12.5V and 15V. However, there was an impact on adjacent healthy tissues, for example, at a control point located in granular tissue, 5 mm from the tumor, where the Arrhenius integral showed a 60% probability of tissue destruction.

The use of techniques aimed at controlling RFA is well-explored in the literature. [31] investigated the effects of pulsating heat on thermal energy transfer in tumor-affected tissue subjected to hyperthermia. The study presented relevant results for the improvement of thermal ablation devices, avoiding temperature peaks and showing that tissue damage related to pulsating heat allows carbonizing the same tumor area as when using a non-pulsating heat source. In this scenario, [32] conducted a study to investigate the effects of different antenna arrangements on the thermal ablation of tumor tissue. Single, double, and triple antenna arrangements were analyzed. The results depict that the use of multiple antennas provides conditions for the expansion of ablation zones. Furthermore, it could be concluded that the use of multiple antennas resulted in lower maximum temperatures compared to simple arrangements.

To investigate the applications of roll-off time delay techniques, careful control of tissue impedance is paramount, whereby the RFA equipment modulates the output power in response to the designated set point established in the targeted tissue, which is diligently monitored throughout the procedure [33]. Manual temperature control of tissue involves maintaining a constant output voltage applied at the electrode tip and adjusting it manually through trial and error to ensure the temperature of the electrode tip remains below the predefined temperature threshold. It is a labor-intensive and time-consuming process that heavily relies on the operator's expertise [34]. On the other hand, automatic temperature control implemented in some RF generators offers a more advanced and reliable approach by utilizing temperature sensors and closed-loop control algorithms to maintain precise temperature control during the ablation procedure.

## Related works

Haemmerich *et al.* [35] reported that temperatures exceeding 50 ˚C induce intracellular protein denaturation and tumor cell membrane destruction, culminating in coagulative necrosis and consequent cell death [36]. Although tissue impedance decreases as heating begins, the dehydration of the target region caused by the vaporization of moisture leads to an increase in impedance [19]. As a result of tissue carbonization, the expansion of the coagulation volume is limited, which compromises the effectiveness of the RFA procedure and exacerbates the roll-off phenomenon [37].

This phenomenon is closely associated with the increase in tissue temperatures, typically reaching 100 ˚C, leading to the fluctuation of tissue impedance throughout the RFA procedure [13]. Roll-off is typically observed during the ablation, after the tissue has been fully desiccated, and the flow and power of the generator have been reduced to zero [38]. Impedance gradual

decrease is observed at the beginning of the RFA procedure. As the tissue temperature increases above 50˚C and goes through the process of carbonization, impedance suddenly rises to values around 1000 Ω [21]. Consequently, the intervention is interrupted due to the inability to effectively conduct an electric current through the tissue [38].

Various strategies have been developed to delay the onset of roll-off, thereby prolonging the duration of RFA [14]. Alternative techniques have been developed to increase the size of coagulation zones, such as the use of internally cooled electrodes in conjunction with the infusion of a refrigerated saline solution [39]. The combined effect of increased tissue conductivity due to the presence of ions from the saline solution and the cooling of the ablation zone by hydration of the target region allows for impedance to be maintained at a reduced level for a prolonged period, thereby enhancing the efficiency of the RFA procedure [19].

In a similar vein, Trujillo and colleagues [13] conducted research utilizing both a theoretical model based on the finite element method and an ex-vivo experimental study. By utilizing porcine liver tissue, they were able to determine that the onset of roll-off is directly linked to the moment when the tissue reaches a point of severe dehydration. Their study demonstrated that infusion of a saline solution can effectively delay the onset of tissue carbonization, thereby postponing the occurrence of roll-off.

Global optimization is a field of applied mathematics and numerical analysis that is specifically concerned with minimizing and maximizing parameters in order to achieve the best possible values for meeting the global objective, while satisfying a set of mathematical modeling criteria, known as the objective function [40]. Bioinspired optimization stands out as one of the most promising methodologies for global optimization. It comprises a set of techniques in the field of computer science that rely on nature-inspired principles for the search of the global optimum. By drawing inspiration from biology and the natural world, bioinspired algorithms have been shown to achieve excellent results in a wide range of optimization problems [41].

The PSO can be applied in various industrial robotic applications. A study conducted by [42] synthesized research related to the motion control of robotic manipulators, with a focus on Super-Twisting Sliding Mode Control techniques. The article highlights the use of optimization algorithms, such as PSO, for an effective approach to tuning these control parameters [42]. Other techniques, like Active Input-Output Sliding Mode Control (AIOFL), are also employed for similar purposes in nonlinear systems. AIOFL is an approach that cancels system disturbances in real-time, transforming it into a chain of integrators up to the relative degree of the system, requiring only knowledge of that degree [43].

In the same context, a study conducted by [44] utilized the PSO algorithm to tune a proportional-integral-derivative (PID) controller for controlling the speed of a permanent magnet direct current motor. Through experiments conducted at an educational level, the authors demonstrated the importance of the number of iterations for the algorithm's performance, which, for this plant and experimental setup, was 25 iterations. Overall, the study yielded promising results in terms of the desired dynamic response of the system concerning rise time, settling time, and overshoot control. The authors conclude this study with the expectation that this work may facilitate the efforts of young researchers and engineers interested in this field.

In the realm of potential optimization algorithms, the Bat Swarm has also been applied in robotics with the aim of finding the shortest path between the initial and final points while avoiding dynamic obstacles. An adaptation of the frequency parameter of the standard Bat algorithm was made to create an enhanced version called the Modified Frequency Bat Algorithm (MFB) [45]. The MFB operates in two modes: first, it generates the trajectory when there are no obstacles in the environment, and second, it avoids obstacles as soon as they are detected. Extensive simulations validated the effectiveness of the MFB, showing that it outperforms the standard Bat algorithm, finding shorter and collision-free trajectories [45, 46].

Research conducted by [47] reveals that the nonlinear fractional-order neural network-based controller (NNFOPID) is designed using nonlinear activation functions in hidden layers and linear functions in the output layers of a neural network, and its tuning is performed through a hybrid optimization algorithm called MAPSO-EFFO. Simulation results show that the NNFO-PID surpasses a neural nonlinear PID controller (NNPID) in terms of trajectory tracking, minimizing mean square error, and reducing energy consumption during circular, linear, and lemniscate trajectories.

In the landscape of cancer research, particularly focusing on liver tumors, this paper distinguishes itself by addressing the limitations of RFA therapies through a comprehensive exploration of diverse techniques. While other studies have primarily concentrated on the thermal aspects of RFA, our work stands out by underscoring the critical role of tissue impedance control as a pivotal parameter in ensuring continuous and successful RFA. Furthermore, the integration of a PSO-tuned PID controller in our methodology demonstrates a novel approach to achieve precise and efficient control, setting our study apart from existing literature. By optimizing the PID controller with the PSO algorithm, we obtain robust results, showcasing low overshoot, rapid rise time, and efficient settling time, thereby contributing to the development of more effective RFA procedures. This unique combination of techniques and emphasis on impedance control positions our work as a valuable contribution to the advancement of liver tumor treatment strategies.

## Roll-off and coagulation zone

The analysis of the roll-off phenomenon in RF procedures is crucial to understand its correlation with the coagulated volume and investigate its potential connection with lesion recurrence. Hypotheses suggest that delaying the roll-off may lead to larger coagulation zones [13, 37]. In a study conducted by Arata *et al* [38], the observation of roll-off occurrence served as a significant predictor of local control after the procedure. Among 20 treated hepatic lesions, tissues that reached the roll-off exhibited a lower rate of local recurrence at 6 months (15%).

To address the issue of roll-off in RF procedures, several approaches have been proposed, including modifications to the mode of energy delivery to the tissue (continuous or pulsed), the use of feedback control laws employing a PID controller [48–50], real-time feedback information (such as tissue temperature, temperature at the active tip of the catheter, or impedance between the active tip and the grounding site), and adjustments for changes in tissue characteristics, such as dehydration. These approaches aim to maintain optimal conditions for the ablation process and avoid the negative consequences associated with roll-off, such as limited expansion of the coagulation volume and compromised therapeutic efficacy [51–53].

Temperature control is a critical aspect of many thermal therapies, and the PID controller is a commonly used method for achieving precise temperature control [51, 54]. This controller is widely utilized due to its ability to provide continuous output variation and high process precision. The PID controller works by measuring the error between the desired and actual temperature and modulating the input voltage in the circuit until the desired output temperature is achieved. It is a simple and easy-to-use control method that has found applications in a wide range of fields [14, 55].

Previous investigations have not incorporated a numerical model for tissue impedance control based on ex-vivo tests using system identification techniques to construct a data-driven dynamic model and apply a tuned controller using the PSO algorithm. The use of the PID controller stands out as a key strategy to mitigate the negative effects of the roll-off, enabling a more precise dynamic control of temperature during the RFA procedure. The implementation

of swarm-based optimization techniques, such as the Particle Swarm Optimization (PSO) algorithm, to adjust the parameters of the PID aims to optimize the system's performance.

The primary objective of this study is to establish dynamic control of tissue impedance for the RFA procedure by implementing a closed-loop PID controller. This control strategy aims to delay the occurrence of roll-off and enhance the overall effectiveness of the intervention. This approach involves adjusting the PID controller to an optimal set of tuning parameters, including proportional gain ($K_p$), integral gain ($K_i$), and derivative gain ($K_d$). The optimization process is carried out using swarm intelligence-based techniques, specifically the PSO algorithm. By iteratively exploring and evaluating potential solutions, the PSO algorithm seeks to identify the most favorable combination of controller coefficients.

Compared to traditional methods like Ziegler-Nichols, this swarm intelligence-based optimization approach offers improved performance and superior results. It leverages the power of intelligent algorithms to fine-tune the PID controller, resulting in enhanced control over tissue impedance during the RFA procedure.

To construct the impedance curve model, ex-vivo experimental data obtained through system identification techniques is utilized. The resulting transfer function, which relates the input voltage (V) to the output impedance ($\Omega$), is carefully analyzed and characterized. This data-driven model forms the basis for the dynamic control of tissue impedance in the RFA procedure.

This study addresses a gap in the literature by focusing on the lack of investigations into roll-off displacement with precise automatic adjustment control using bioinspired computational techniques such as PSO. Additionally, it aims to enable deployment of this control on hardware for real-time execution. In future work, we plan to integrate a dynamic actuation controller. This controller will receive information on tissue impedance variation and simultaneously adjust the PID controller to displace the roll-off, thus providing a more efficient ablation procedure.

## Materials and methods

The methodology of the study is graphically presented in Fig 1, which summarizes all the steps involved in the optimization process.

### Experimental protocolol

The experimental procedure utilized ex-vivo porcine liver tissue that was sourced from local markets and maintained at temperatures ranging from 18˚C to 22˚C. The organs were sectioned into cubes measuring approximately 6 cm on each side. The RFA equipment utilized in the experiment was developed by the Biomedical Engineering Laboratory (LaB) at the

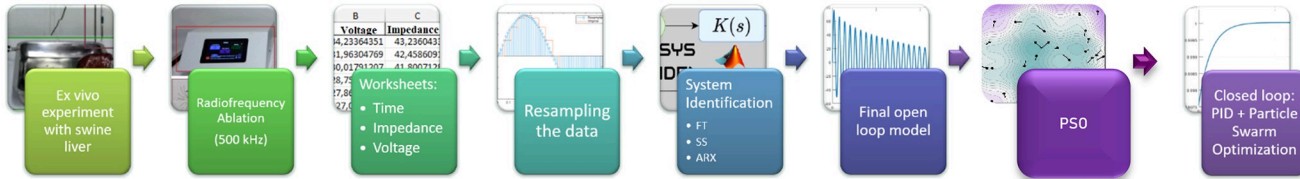

**Fig 1. Methodological flow of theoretical-experimental procedure.** Initially, we conducted an ex-vivo experiment using porcine liver tissue, which was subjected to RFA using the SOFIA® equipment. The obtained data were organized in spreadsheets and resized. Next, SI was performed using the ARX method, and the model was converted to the TF domain. With the open-loop system, the PSO algorithm was applied (Fig 2), returning optimal gains for use in the PID controller.

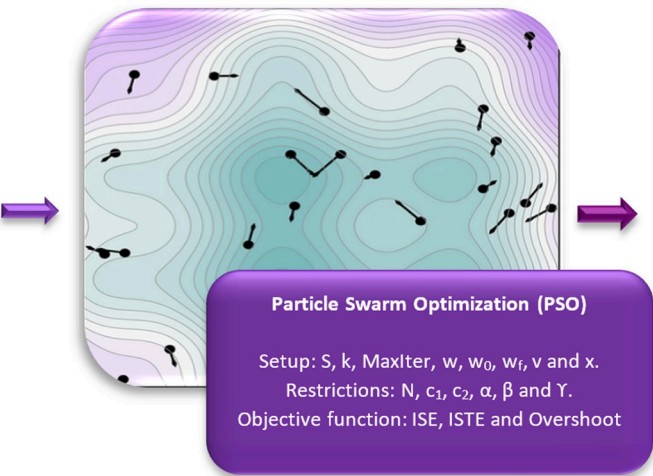

**Fig 2. Expanded PSO (Fig 1).** In the open-loop system, the PSO algorithm was utilized, incorporating variable parameters such as the number of particles (S), iteration index (k), the maximum number of iterations (MaxIter), inertia factors calculated per iteration, initial and final ($\omega$, $\omega_0$, $\omega_f$), and search intervals covering both position and velocity (x and v). On the other hand, fixed parameters, including the number of dimensions (N), cognitive and social coefficients ($c_1$ and $c_2$), as well as the weighting values of performance indicators $\alpha$, $\beta$, and $\gamma$, remained constant throughout the process and were assessed through the objective functions of the integral of the squared error criterion (ISE), integral of the squared error weighted over time (ISTE), and overshoot. As a result, the achievement of optimal gains that can be employed in the PID controller was demonstrated.

University of Brasilia (UnB). The equipment, named SOFIA®, has a patent request (BR 10 2017 002683 3) [56].

The Boston Scientific LeVeen® Standard 4.0 umbrella electrode (Marlborough, MA, USA) with a diameter of 2.5 cm was employed in all trials with the electrode in the semi-open position, following the RFA test protocol of the SOFIA® equipment [57, 58]. All components of the ex-vivo experiments are shown in Fig 3.

The test was performed with an initial power of approximately 34 W applied by the monopolar electrode on the liver samples. The dispersive electrode was connected to an aluminum base covered by an acrylic plate, allowing only the section of the same dimensions as the liver sample to touch the ground terminal (see Fig 3E). The stopping criterion used was the detection of the first roll-off reached by the tissue, measured by the SOFIA® equipment. Maintaining a constant power level was necessary to ensure experiment reproducibility.

The temperature was measured using a thermocouple sensor provided with the SOFIA® equipment. The sensor was placed at a distance of 1.25 cm from the center of the electrode, located at the tip of one of the twelve electrode rods forming an "umbrella" geometry (see Fig 3C).

The VERA system, a continuous monitoring system connected to SOFIA® via serial communication, was used to monitor the parameters of the RFA signal. The system has a patent request (BR 10 2017 002919 0) [59].

## System identification—Roll off

The application SI techniques to the dynamic modeling of the roll-off phenomenon enables the correlation of the applied voltage in the RFA procedure with the behavior of tissue impedance.

**Data acquisition.** The open-loop SI is established by evaluating the input values *u* and output *y*. In this study, a 500 kHz sinusoidal voltage was utilized as input data, with effective

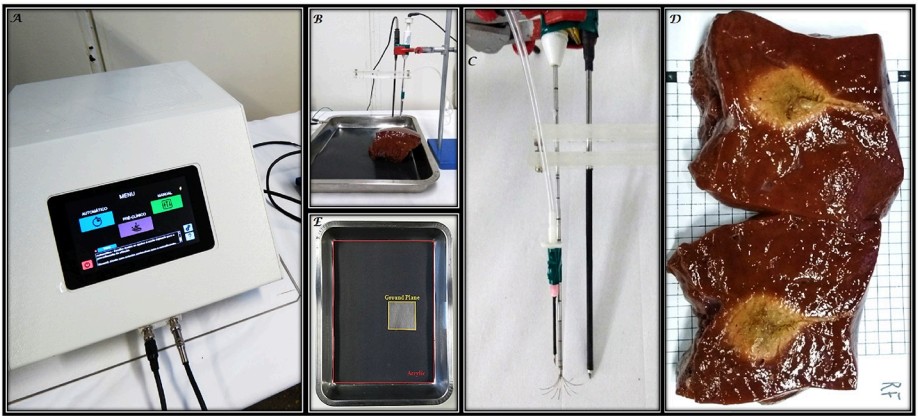

**Fig 3. Experimental setup for ex-vivo tests.** Fig (A) shows the RFA equipment SOFIA® developed by the Biomedical Engineering Laboratory of the University of Brasília, adjusted to deliver an initial power of approximately 34 W through a 500 kHz sine wave electrical current. Fig (B) depicts the positioning of the piece of swine liver on the bench where the tray is connected to the dispersive electrode. In Fig (C) we have the Boston Scientific LeVeen Standard 4.0 umbrella electrode with a diameter of 2.5 cm responsible for supplying high-frequency energy for the procedure and a thermocouple sensor placed at a distance of 1.25 cm from the center of the electrode. In (D), we present the result of the RFA in one of the 6 cm cubic pieces of liver, which is sectioned in the middle, showing the region of hepatic tissue necrosis. In (E) we have the grounding plane that was connected to the dispersive electrode assembled from an aluminum tray coated with a layer of acrylic. A 6 cm x 6 cm window was cut out so that the pieces of hepatic tissue remained in contact with the grounding surface in a region with the same dimensions as the cubes. All procedures were based on the SOFIA® testing protocol, which can be consulted in references [57, 58].

RMS values measured in volts. The output data represents the impedance of the liver tissue, calculated from the voltage and current measurements obtained via the SOFIA® equipment. Only the real part of the impedance was considered as it is associated with Joule heating [60]. Reactances were neglected.

**Model structure.** The data generated by the SOFIA® equipment were exported to MATLAB® R2021a (MathWorks, Natick, MA) for analysis. In order to overcome any data collection flaws, we conducted multiple experiments and selected one that exhibited measurements with the highest suitability for SI. Despite selecting the optimal experiment, the SOFIA® equipment exported measurements that did not possess constant time intervals $\Delta t$. Therefore, upon importing the data into MATLAB®, we utilized the resample function, which through data interpolation, eliminated the temporal gaps present in the measurement process, thus achieving uniformity in the voltage and impedance dataset with respect to $\Delta t$. The data was resampled at a rate of 1 measurement per second.

The acquired data was processed through the "iddata" command in MATLAB®, which enabled the creation of an object encapsulating the input/output data along with their respective properties such as labels, units of measure, and initial time. Subsequently, the System Identification Toolbox was utilized to load the data object into the environment where the identification of the system was performed.

After importing the data, we proceeded to identify discrete-time and continuous-time transfer function (TF) and state-space models of $2^{nd}$, $3^{rd}$, $4^{th}$, $5^{th}$, and $6^{th}$ order using the System Identification Toolbox in MATLAB®. To obtain the parameters for these models, we employed the prediction error minimization (PEM) method without perturbations. Although the results obtained were not satisfactory.

To process the data and obtain a more accurate model curve that closely fits the experimental data, we developed a MATLAB® script based on the autoregressive representation with

exogenous inputs (ARX), which models the system as a linear combination of past outputs, past inputs, and system noise. This approach assumes that the data were collected at defined instants of time and allows us to mathematically represent the system using (1) [61]:

$$A(z^{-1})y(k) = B(z^{-1})u(k) + \xi(k) \tag{1}$$

Where $(k = 1, 2, \ldots, N)$, $y(k)$ represents the output signal at time instant $k$, $u(k)$ represents the input signal at time instant $k$, $\xi(k)$ represents the system noise (error in the model or in the measurements) at the time instant $k$ and $A(z^{-1})$ e $B(z^{-1})$ are polynomials in the backward shift operator $z^{-1}$, representing the system dynamics:

- $A(z^{-1}) = 1 + a_1 z^{-1} + \ldots + a_{na} z^{-na}$;
- $B(z^{-1}) = 1 + b_1 z^{-1} + \ldots + b_{nb} z^{-nb}$;

where $na$, $nb$ are non-negative integers indicating the order of the model.

From (1), we obtain the linear regression model (2):

$$\mathbf{y} = \varphi \boldsymbol{\theta} + \boldsymbol{\xi} \tag{2}$$

Where $u(k)$ and $y(k)$ are the sampled observations from the inputs and outputs of the system, respectively. $\varphi$ is the regression matrix, with dimensions $(N - p + 1) \times (na + nb)$, $\theta$ are the model parameters, $\xi$ is a vector of residues, distributed in an independent, uniform way, with zero average and finite variance and $p = 1 + \max(na, nb)$. We present the expansion of the terms in (2) in sequence:

$$\varphi(k) = \begin{bmatrix} -y(k-1) \\ -y(k-2) \\ . \\ . \\ . \\ -y(k-na) \\ u(k-1) \\ u(k-2) \\ . \\ . \\ . \\ u(k-nb) \end{bmatrix}, \theta = \begin{bmatrix} a_1 \\ a_2 \\ . \\ . \\ . \\ a_{na} \\ b_1 \\ b_2 \\ . \\ . \\ . \\ b_{nb} \end{bmatrix}$$

$$\begin{bmatrix} y(p) \\ y(p+1) \\ . \\ . \\ . \\ y(N) \end{bmatrix} = \begin{bmatrix} \phi^T(p) \\ \phi^T(p+1) \\ . \\ . \\ . \\ \phi^T(N) \end{bmatrix} \theta + \begin{bmatrix} \xi(p) \\ \xi(p+1) \\ . \\ . \\ . \\ \xi(N) \end{bmatrix}$$

To estimate the unknown parameters of the vector $\theta$ referring to the coefficients of the polynomials $A(z^{-1})$ and $B(z^{-1})$, we employed the least squares algorithm (3):

$$\hat{\boldsymbol{\theta}} = (\varphi^T \varphi)^{-1} \varphi^T \mathbf{y} \tag{3}$$

Once the SI was performed to depict the roll-off, the dataset generated using the ARX model with one-step forward prediction and $na = 20$ and $nb = 20$ (4) was converted to the discrete-time TF domain:

$$y(k) = -a_1 y(k-1) - a_2 y(k-2) \cdots - a_{20} y(k-20)$$
$$+ b_1 u(k-1) + b_2 u(k-2) \cdots + b_{20} u(k-20) \tag{4}$$

We utilized the "tfest" command in MATLAB® to obtain the discrete-time TF model from the dataset. The initial model fit was retained, and subsequently, the model was converted from discrete to continuous time domain for utilization in the PSO algorithm for PID controller design.

**Validation.** The model validation procedure was performed based on the data obtained from the experiments conducted. The experiments were partitioned into two subsets: one for SI and the other for model validation. Initially, we assumed that all models were adequate. To determine the best experiment to include in the validation dataset, we conducted a thorough evaluation of the quality of fit by analyzing the observation and estimation data in both time and frequency domains, as well as analyzing residual plots and reliability regions.

To assess the robustness of the models to experimental data, we employed the "Fit to estimation data" (FIT) index, which is a performance measure commonly used to evaluate the quality of fit between the estimated and reference data for the analysis and validation of the identified models [62]. The FIT index was computed using the "Normalized root mean squared error" (NRMSE), which provides a quantitative measure of the accuracy of the model predictions (see 5):

$$FIT = \frac{\sqrt{\sum_{k=1}^{N} [\xi(k)]^2}}{\sqrt{\sum_{k=1}^{N} [y(k) - \bar{y}]^2}} \cdot 100 \tag{5}$$

In this context, we also considered the coefficient of determination $R^2$ (6):

$$R^2 = 1 - \frac{\sum_{k=1}^{N} [\xi(k)]^2}{\sum_{k=1}^{N} [y(k) - \bar{y}]^2} \tag{6}$$

Where $y$ and $\bar{y}$ are the output data from the experiments and the mean of the experiments, respectively. This is a statistical measure of how well the estimated model fits the prediction data. This metric quantifies the degree of correlation between the estimated and predicted values and is commonly used to evaluate the accuracy of the identified model.

## Control design

**PID controller.** To prevent roll-off, we propose incorporating a PID controller into the SI. The PID controller is a control technique that combines proportional, integral, and derivative actions to improve the dynamic response of a system. In our impedance control protocol, we use a closed control loop with negative feedback to measure the error, which is the difference between the current tissue impedance (in ohms) and the desired impedance (set point).

The controller takes the error as input and modulates an output voltage (in volts) applied between the active electrodes and the ground terminal using PID actions. Generally, the control loop is composed of two parts, in terms of TF relating the input and output signals: the controller itself, given in (7), and the system plant, given in (16). The plant is obtained from the SI and has the voltage modulated by the controller as input and the tissue impedance as output:

$$K(s) = K_p + \frac{K_i}{s} + K_d s \tag{7}$$

The modulated output voltage that is applied to the electrode tip in the continuous time domain can be represented by (8) [34]:

$$v(t) = K_p e(t) + K_i \int_0^t e(\tau) d\tau + K_d \frac{d}{dt} e(t) \tag{8}$$

In this work, we utilize the quadratic error integral criterion (9):

$$ISE = \int_0^\infty e^2(t) dt \tag{9}$$

and the time-weighted quadratic error integral (10):

$$ISTE = \int_0^\infty t e^2(t) dt \tag{10}$$

These serve as performance measures to assess the effectiveness of the PID controller. These measures are assigned to the reduction of overshoot and accommodation time of the manipulated variable to the set point, respectively.

We determined the proportional $K_p$, integral $K_i$, and derivative $K_d$ gains using the evolutionary PSO algorithm. In this context, the objective function (OF) we adopted was based on the linear combination of performance parameters in the continuous time domain.

To obtain performance indicators that provide good dynamic responses, we aimed to minimize the OF, given by (11):

$$argmin(OF = \alpha ISE + \beta SysOvr + \gamma ISTE) \tag{11}$$

Where the *ISE*, the overshoot of the system (*SysOvr*) and the *ISTE* are parameters obtained from SIMULINK®. The values $\alpha$, $\beta$ and $\gamma$ are weights referring to each of the performance indicators that should be reduced. Thus, according to the behavior of the system to be controlled, the weights were varied in order to point out to the algorithm which indicator needs more attention to obtain the optimal gains. The block diagram of the circuit designed in SIMULINK® is shown in Fig 4.

The stability analysis of the system is crucial to ensure the effectiveness of the controller design. To this end, we employed the Routh-Hurwitz criterion to analyze the stability of the system, represented by the TF obtained from SI and the PID controller. Specifically, we evaluated the characteristic polynomials and the distribution of the roots in the complex plane to ensure stability.

**Particle swarm optimization.** PID controllers are widely used to handle applications that require high precision, such as motion control, positioning, and heat treatment [63, 64]. Various controller tuning methodologies have been developed to achieve optimal performance, and among these, the Ziegler-Nichols method is considered to be one of the most well-known techniques [65, 66]. However, determining the optimal or near-optimal gains using the Ziegler-Nichols formula has a tendency to generate a large overshoot [67].

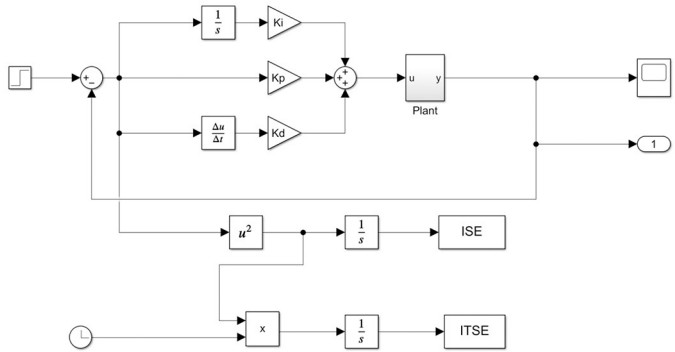

**Fig 4. Block diagram of the circuit designed in SIMULINK®.** The block diagram highlights the circuit designed in Simulink software for determining the gains $K_p$, $K_i$, and $K_d$ of the PID controller. Additionally, it shows the presence of the ISE and ISTE errors, which are calculated and sent to the script in MATLAB® software to compose the OF. Also visible are the presence of integrator blocks $\frac{1}{s}$, differentiator $\frac{\Delta u}{\Delta t}$, the system plant highlighted in 16, blocks performing quadratic calculations ($u^2$) and product ($x$) of variables besides the output variable (1) that returns values to the PSO algorithm for calculating the overshoot and being added to the OF.

In this work, we use the PSO algorithm to tune the PID controller. The PSO algorithm was proposed by Eberhart and Kennedy [68] and is based on computational search and optimization techniques inspired by the collective intelligence of swarms, schools of fish, and the social behavior of birds when searching for food [69].

Each particle in the PSO algorithm is treated as a zero-volume element in an N-dimensional space [67], where N represents the number of tuning parameters of the controller. The particles work collaboratively and competitively to find the optimal solution, with each particle representing a possible potential solution [70, 71]. Each particle adjusts its search direction based on its own experience and the knowledge of its fellow swarm particles. It is attracted to the best solution that any particle in its neighborhood has found [65].

In this sense, the swarm is initiated by randomly distributing particles in the search space, represented by position and velocity vectors [72]. The $i^{th}$ particle is then defined by three vectors:

- The coordinates of an individual particle in the N-dimentional search space
$\vec{x}_i = (x_{i,1}, x_{i,2}, \cdots, x_{i,N})$;

- The rate of change in position (velocity)
$\vec{v}_i = (v_{i,1}, v_{i,2}, \cdots, v_{i,N})$;

- The position of best fitness that each particle has achieved (*pbest*)
$\vec{p}_i = (p_{i,1}, p_{i,2}, \cdots, p_{i,N})$.

The search for an optimal solution in the PSO algorithm involves determining the best overall fitness position, considering all elements globally. This is accomplished through the variable gbest. The velocity and position updates of each particle are then calculated based on the current velocity and the distance between the particle's personal best position (pbest) and the global best position (gbest). In an examination involving $S$ particles, the position of the $i^{th}$ particle in the $j^{th}$ dimension is updated using (12) and (13):

$$v_{i,j}^{k+1} = \omega v_{i,j}^{k} + c_1 r_1 (pbest_{i,j} - x_{i,j}^{k}) + c_2 r_2 (gbest - x_{i,j}^{k}) \tag{12}$$

$$x_{i,j}^{k+1} = x_{i,j}^{k} + v_{i,j}^{k+1} \tag{13}$$

Where the iteration index is denoted by *k*, while the maximum number of iterations is represented by the variable MaxIter, which is used as a stopping criterion. Specifically, the algorithm continues to execute while the condition $0 < k \leq MaxIter$ is satisfied.

The values of $c_1$ and $c_2$ correspond to the cognitive and social coefficients, respectively, and are related to the degree of self-confidence and trust within the swarm. These values determine the influence of the particle's own experience and the collective knowledge of the swarm on its movement. They are usually set to constant values, although adaptive versions of PSO exist that dynamically adjust these coefficients during the optimization process. Additionally, the values of the maximum and minimum velocities ($V_{max}$ and $V_{min}$, respectively) and positions ($X_{max}$ and $X_{min}$, respectively) are pre-determined and constrain the particle's movements to remain within the search space.

The values of $r_1$ and $r_2$ are random numbers that are uniformly distributed in the range of 0 to 1. The variable $\omega$ represents the inertia factor, which is used to decrease the magnitude of the velocity during the iterations. The inertia factor is applied in the calculation of the velocity according to (14):

$$\omega^{(k+1)} = \omega^k + \frac{(\omega_f - \omega_0)}{MaxIter} \tag{14}$$

Where $\omega_f$ and $\omega_0$ are the final and initial inertia factors, where we use the values 0.1 and 0.9, respectively.

In other words, the inertia factor is utilized as a scaling parameter to adjust the current velocity of each particle. A proper choice of $\omega$ can strike a balance between global and local exploration, thereby reducing the number of iterations required to find a satisfactory solution [67].

Based on the simulation results, we have identified the parameters mentioned in this section that provide satisfactory performance. To ensure the validity of these parameters, we have conducted several validation tests. Our parameter selection was guided by research works cited in this paper, including [65, 67, 72–76]. The mentioned articles provide valuable information on PSO algorithm optimization, including the selection of appropriate values for its parameters based on swarm communication topologies and proposed modifications to the original algorithm. The flowchart has been appropriately included in the supporting information (see Appendix A in S1 Fig).

## Results

Despite being widely used for the treatment of HCC, RFA still requires parameter optimization and techniques that can provide better clinical outcomes, safety, and procedural efficacy. In this study, we demonstrate a System Identification (SI) technique to analyze tissue impedance behavior during RFA procedures through computer simulation. The impedance analysis, associated with the roll-off phenomenon, can result in a more effective procedure by extending the ablation time without damaging adjacent tissues. The data were derived from a previous ex vivo study conducted by our research group. In this study, we employed a PID controller to adjust the occurrence of the roll-off phenomenon using a PSO algorithm.

### Ex vivo data obtained for simulation of parameters

The experiments were conducted with an initial power of approximately 34 W applied by the monopolar electrode to the liver samples. In Fig 5, for the estimation and validation experiment, we can observe that in the first 30 seconds of the ablation procedure, both the voltage and power decrease, stabilizing around 26 V and 16 W, respectively. This event is also

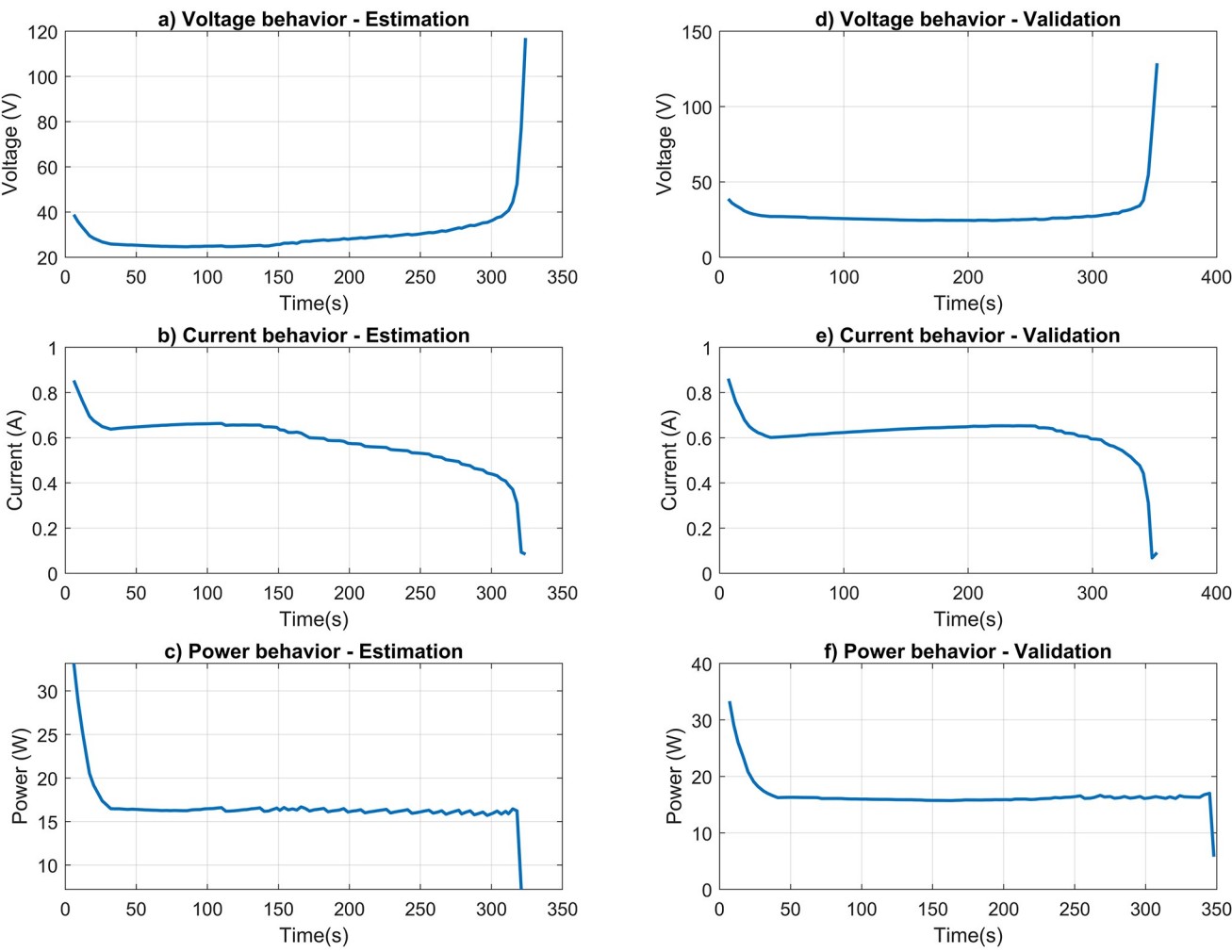

**Fig 5. Voltage, current and power behavior.** The experimental setup involved initiating the procedure with a power of approximately 34 W using a monopolar electrode on the liver samples. Throughout the estimation phase (Figs a, b, and c), the behavior of voltage, current, and power was meticulously observed. The validation phase (Figs d, e, and f) mirrored these observations. The occurrence of roll-off, a crucial phenomenon in the ablation process, was clearly evidenced in both phases. Roll-off is characterized by a significant decrease in current and power, accompanied by a notable increase in voltage, indicating a sharp rise in impedance. This reduction in current and power can be attributed to the principle that the energy transferred per unit of time is directly proportional to the square of the current, highlighting its pivotal role in the procedure.

observed in ex vivo liver experiments by [77]. The power and voltage exhibit this behavior due to the initiation of current circulation in the RF generator, causing a slight drop in power due to the applied load. In this scenario, at the beginning of the ablation procedure, the increase in temperature leads to an increase in the electrical conductivity of the tissue, causing the tissue impedance to drop slightly in the initial moments [78].

As we approach the moment of roll-off, there is a sharp drop in current and power, which can be attributed to the impedance behavior. According to [79], temperatures above 85°C result in a decrease in electrical conductivity due to progressive tissue dehydration. In this context, considering the relationship of Ohm's law, we can infer that the sudden increase in voltage is related to the increase in tissue impedance and decrease in current. The decrease in power is associated with the relationship between power, impedance, and current, where the latter exhibits quadratic behavior in the mathematical expression.

## System identification

Based on the ex-vivo experiments conducted, we selected the most promising assays for use in SI for the estimation and validation phases of the model based on the comparison of obtained roll-off curves.

Firstly, we initiated the SI process using MATLAB® System Identification Toolbox. Several models such as TF, state-space representations, both in discrete and continuous-time domain were applied to the experimental data, however, satisfactory results were not obtained, or the models did not converge. From this point, the next step was to use the ARX polynomial model.

Fig 6 displays the tissue impedance measurements obtained during the estimation and validation phases for the selected experiment at each time instant. In the following, we present the behavior of the data that we used for model estimation and validation.

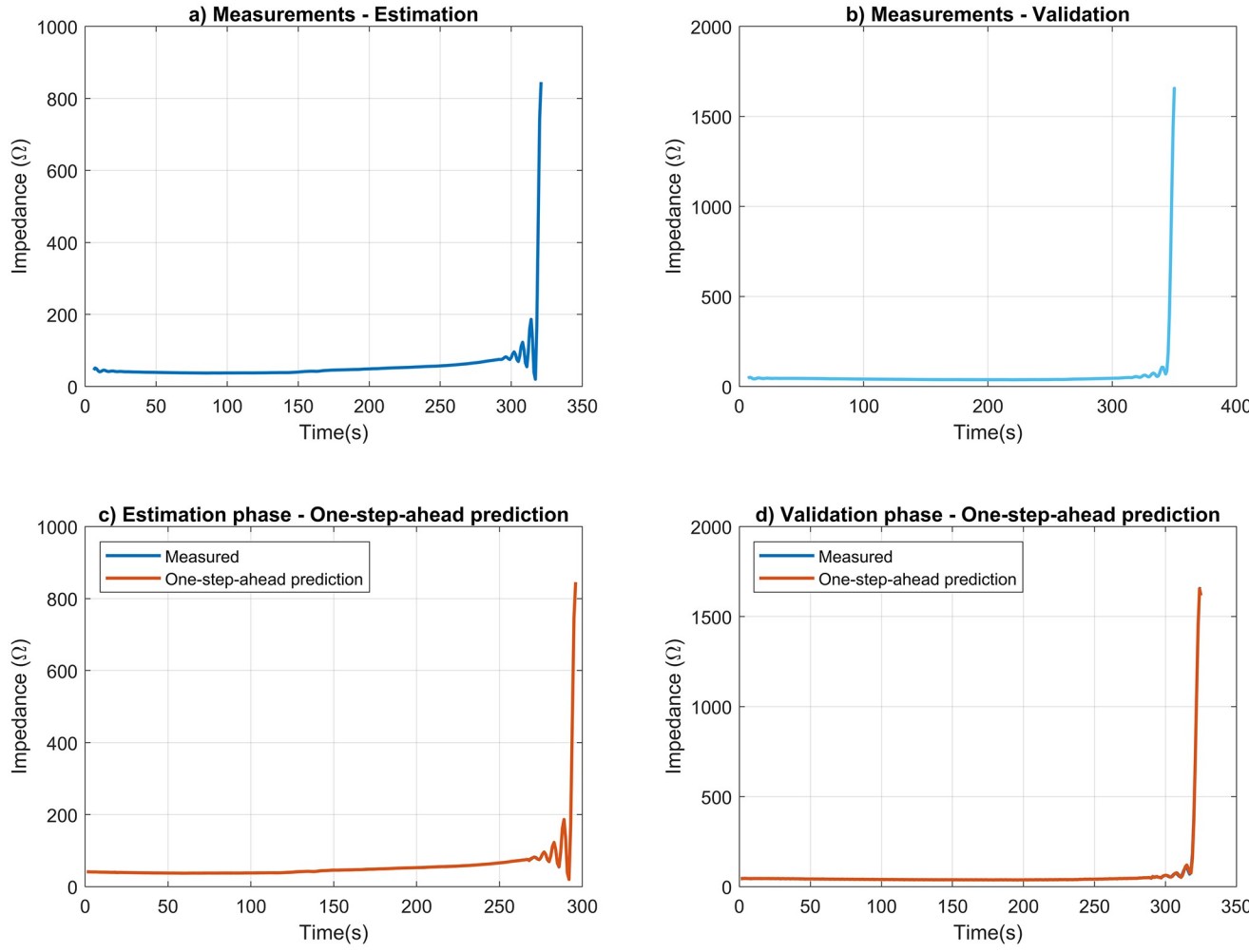

**Fig 6. System identification graphs.** The tissue impedance measurement procedure in this study involved using the ARX model to measure tissue impedance (Ω) over time(s), with separate estimation and validation datasets. The procedure comprised four main steps: (a) displaying tissue impedance measurements related to SI, (b) displaying validation data measurements, (c) estimating phase using measurement and prediction curves one step ahead, and (d) validating the phase using measurement and prediction curves one step ahead. Plots for each step of the SI procedure are presented in corresponding subfigures (a), (b), (c), and (d), providing a detailed overview of the approach. As discussed in the previous subsection, the impedance behavior in both the estimation and validation datasets is clearly observable. Additionally, the graphs indicate that roll-off occurred at distinct time intervals, precisely at 318 s and 345 s, respectively. The prediction curves almost overlap with the measurement data, demonstrating the method's effectiveness for these experiments.

After obtaining the measurements, we employed the SI procedure with a time interval of 316 seconds after resampling the data. During the simulation to determine the order of the ARX model, we observed that higher order polynomials resulted in a better fit. However, when we transformed the model to continuous time TF representation, the FIT index was reduced, resulting in an unsatisfactory degree of accuracy.

Empirically, we determined the optimal orders of the polynomials $A(z^{-1})$ and $B(z^{-1})$ for the ARX model through computer simulations and verification of the FIT indices. We selected $na = 20$ and $nb = 20$ for the discrete-time model.

Fig 6(c) and 6(d) demonstrate the results of the one-step-ahead prediction using the model given in (4) for both the estimation and validation data, respectively, depicting the behavior of the tissue impedance. The FITs for the estimation and validation data were 99.628% and 97.872%, respectively. The correlation ($R^2$) values obtained were 0.99999 and 0.99955 for the estimation and validation data, respectively.

To ensure the adequacy of the ARX model, we conducted an analysis of the statistical properties of the residuals, in addition to selecting the best order of the model based on the fitting results. Fig 7 shows the autocorrelation and cross-correlation ranges of the residuals in relation to the input signal of the model. The plot indicates that the residuals exhibit low levels of autocorrelation and cross-correlation, and the dotted lines represent the threshold limits for adequacy of the model, where estimated residuals are approximately uncorrelated.

The bounded regions indicate the results of statistical validation tests, with 95% confidence, computed using the difference between the experimental impedance data and the one-step-ahead prediction of the input data (voltage).

Fig 7 comprises five representations. The first three show the autocorrelation and cross-correlation analysis from a linear point of view, which is the nature of the identification method used. The next two representations demonstrate an analysis carried out to check for any non-linear correlations of the residuals.

It can be observed that a small portion of the correlation tests for the residuals falls outside the 95 percent confidence margin. This situation is attributed to the nonlinear behavior of the curve that describes the roll-off phenomenon. However, nonlinear tests were performed, but did not yield better FIT results than the adopted model. Additionally, the small extrapolations do not significantly affect the dynamic performance of the model [80].

From (4), we obtain the plant discrete-time TF $P_D(z^{-1})$ (15):

$$P_D(z) = \frac{33.37z^{-1} - 96.47z^{-2} + 120.6z^{-3}}{1 - 2.42z^{-1} + 2.72z^{-2} - 1.79z^{-3} + 0.45z^{-4}} \cdots$$

$$\cdots \frac{-80.08z^{-4} + 20.9z^{-5} + 1.71z^{-6}}{0.37z^{-5} - 0.68z^{-6} + 0.54z^{-7} - 0.17z^{-8}}$$

$$(15)$$

This representation maintained the polynomial model's good FIT. Then, we extracted the continuous-time TF representation to capture the behavior of the roll-off phenomenon, as shown in (16):

$$P(s) = \frac{32.43s^8 + 135s^7 + 510.3s^6 + 925.7s^5}{s^9 + 2.12s^8 + 15.76s^7 + 21.61s^6 + 55.28s^5} \cdots$$

$$\cdots \frac{1104s^4 + 829.7s^3 + 556.9s^2 + 58.83s + 0.81}{46.8s^4 + 49.14s^3 + 24.45s^2 + 7.28s + 0.5}$$

$$(16)$$

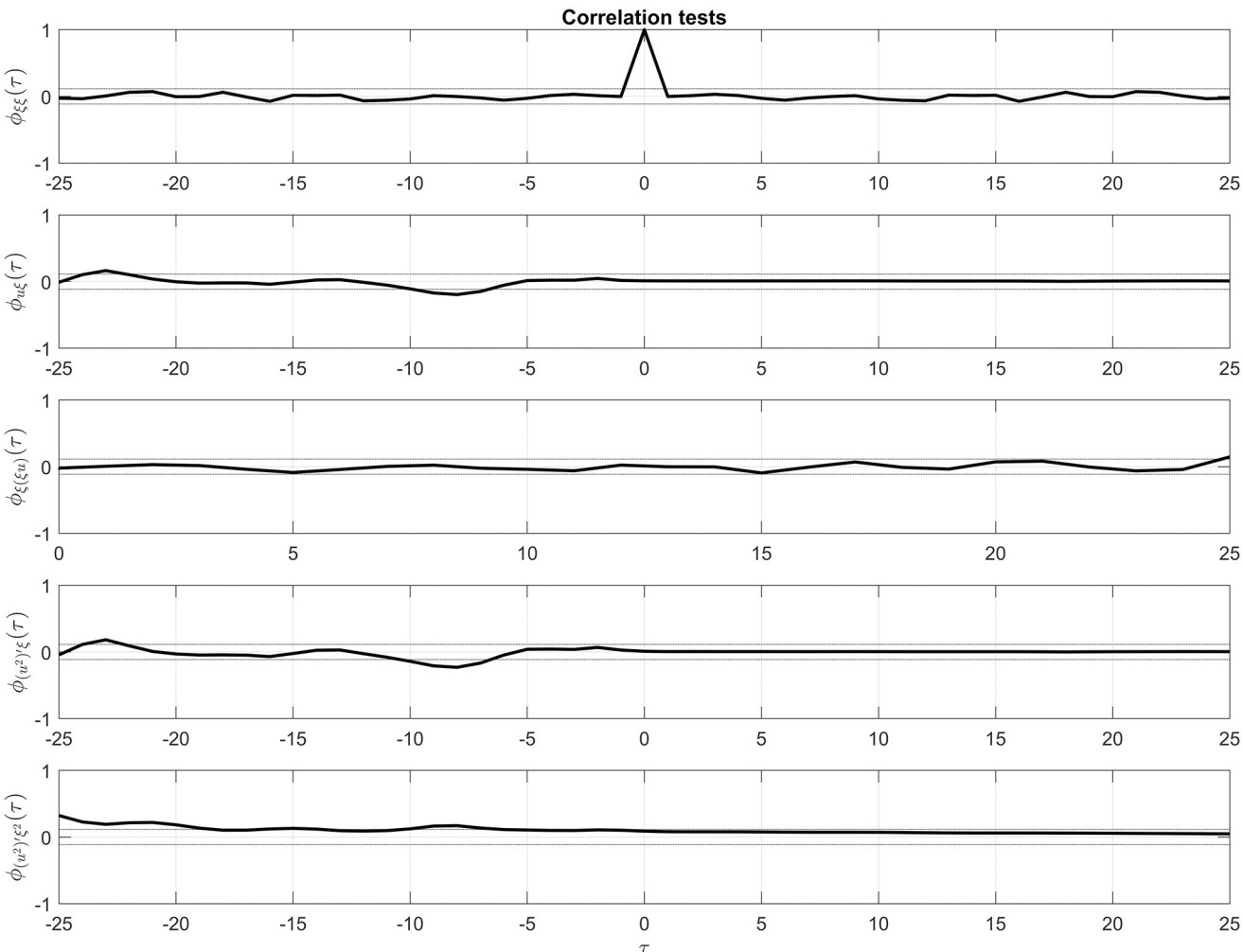

**Fig 7. Correlation tests.** The graph displays the results of correlation tests conducted on the residuals. The first three plots depict the analysis of linear correlation, while the following two plots illustrate the analysis of non-linear correlation and autocorrelation. Dotted lines represent the 95% confidence level limits. It is worth noting that these residual tests are more concerned with the statistical performance of the estimator than with the dynamic performance of the identified system. Additionally, while the ARX method is linear, we employed non-linear correlation analysis due to the non-linear behavior exhibited in the impedance curves in both the estimation and validation phases.

We conducted an empirical estimation of the TF by testing several model orders. The one with 9 poles and 8 zeros resulted in the best FIT, with an index of 85.29% and 64.16% for the estimation and validation data, respectively, as shown in Fig 8.

We proceeded to apply the Routh-Hurwitz criterion to verify the absolute stability of the open-loop system corresponding to the TF. This analysis avoids the need to factorize the characteristic equation polynomial.

Table 1 shows the results of this analysis. The number of sign changes in the first column corresponds to the number of poles located on the right side of the complex plane. If all elements have the same sign, the system is considered stable.

After verifying the stability of the open loop system, we can now analyze the dynamic response of the model. Fig 9 shows the step response and the root locus of the system.

The step response shown in Fig 9 displays a significantly high overshoot of around 1980% for a unit input signal. Furthermore, the open-loop system's dynamic response exhibits a long

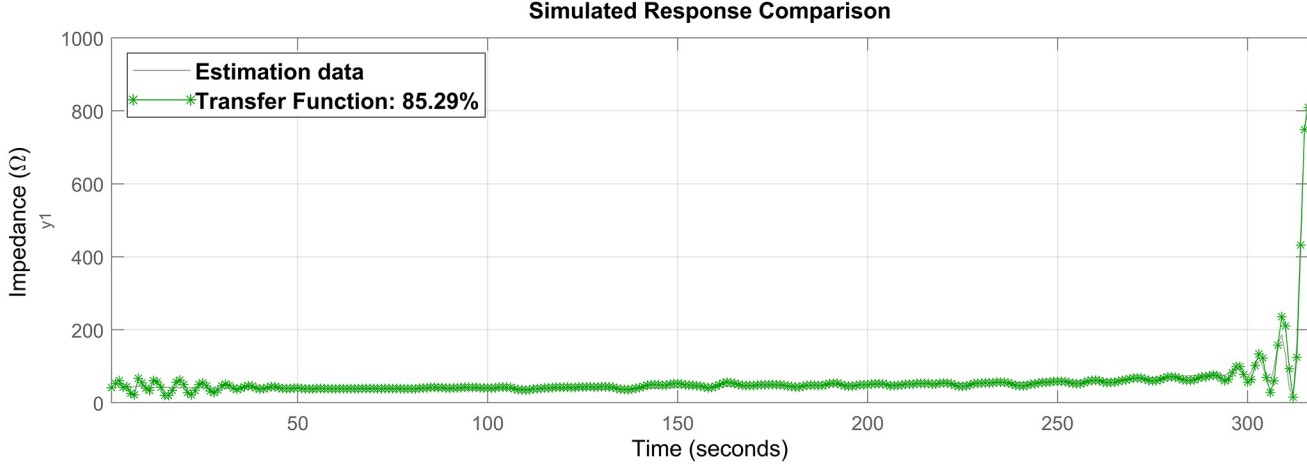

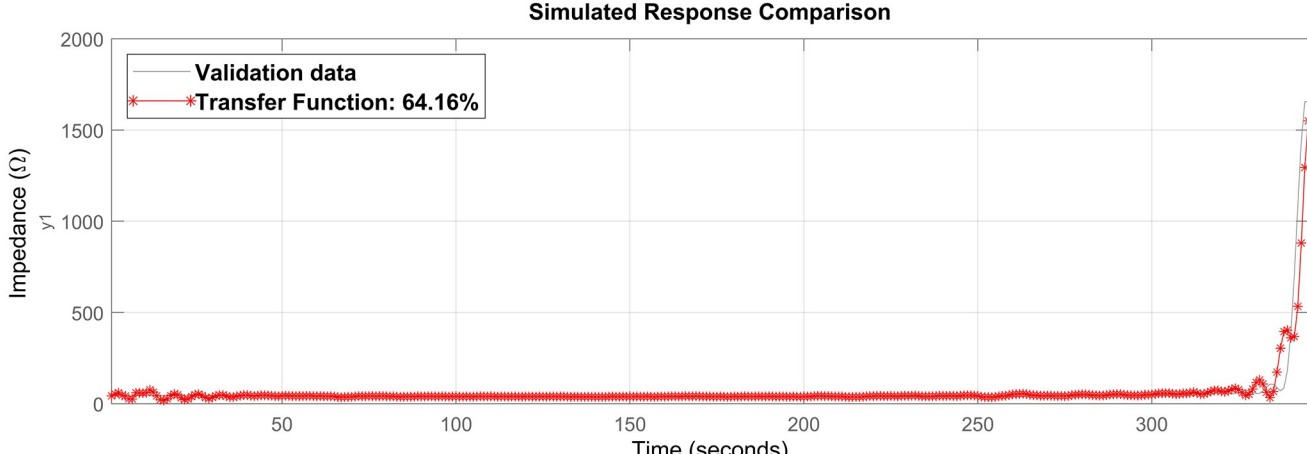

**Fig 8. Transfer function camparison graphs.** These graphs depict the comparison between the data extracted from the ex-vivo experiment and the data belonging to the TF model, as presented in 16. The green curve represents the model fit to the estimation data (85.29%), while the red curve represents the fit to the validation data (64.16%). It is noteworthy that the system identification tests conducted directly from the measured data did not yield satisfactory results. The model fit obtained is due to the use of the ARX method with one-step-ahead prediction, which provided a new set of data from which the TF was derived.

**Table 1. Routh-Hurwitz stability criterion for the $P(s)$ TF from SI: Open-loop roll-off system.**

| | | | | | |
|---|---|---|---|---|---|
| $s^9$ | 1 | 15.7632 | 55.2829 | 49.1387 | 7.2780 |
| $s^8$ | 2.1186 | 21.6147 | 46.7952 | 24.4473 | 0.5032 |
| $s^7$ | 5.5608 | 33.1950 | 37.5992 | 7.0405 | . . . |
| $s^6$ | 8.9679 | 32.4704 | 21.7650 | 0.5032 | . . . |
| $s^5$ | 13.0608 | 24.1033 | 6.7285 | . . . | . . . |
| $s^4$ | 15.9204 | 17.1450 | 0.5032 | . . . | . . . |
| $s^3$ | 10.0379 | 6.3156 | . . . | . . . | . . . |
| $s^2$ | 7.1282 | 0.5032 | . . . | . . . | . . . |
| $s^1$ | 5.6070 | . . . | . . . | . . . | . . . |
| $s^0$ | 0.5032 | . . . | . . . | . . . | . . . |

In summary, the Routh-Hurwitz criterion provides a useful tool for determining the stability of a TF obtained from SI, allowing for the analysis of its dynamic behavior and identification of the need for a controller to ensure stability.

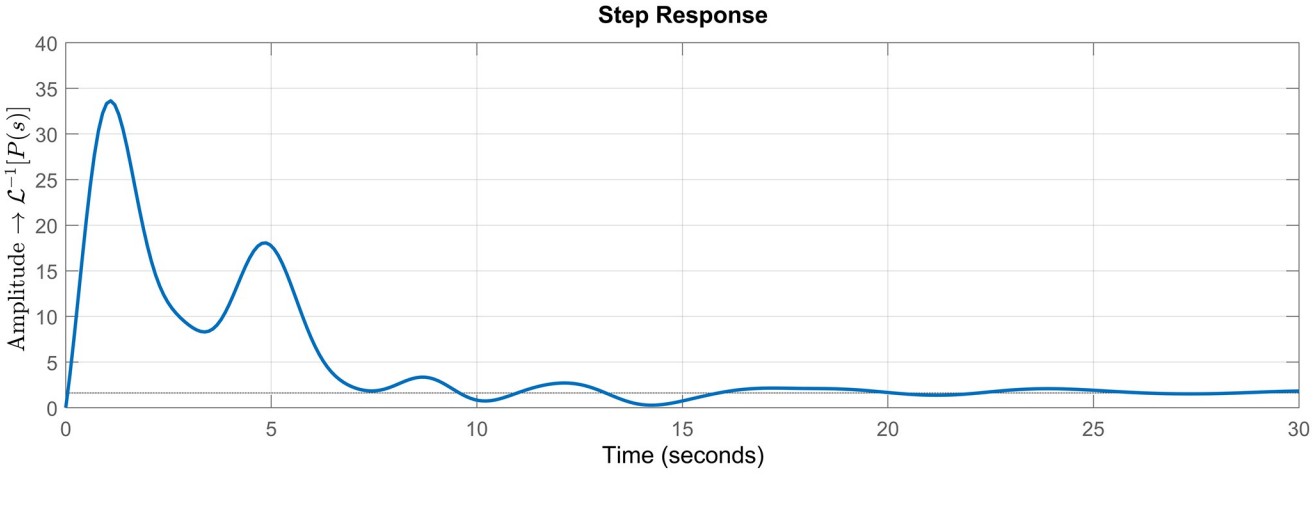

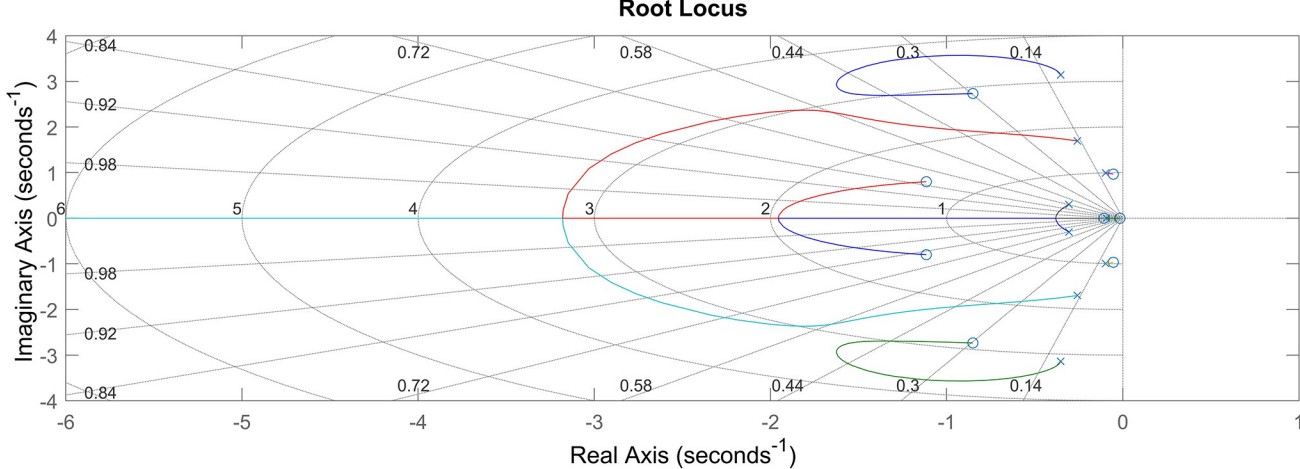

**Fig 9. Step response and root locus.** The dynamic step response of the open-loop model, where the input is voltage and the output is impedance, is presented. Along with the root locus plot, the figures demonstrate that the system response is composed of several terms that promote oscillatory behavior and does not reach the desired setpoint. The step response exhibits a rise time of 0.0364 s, a peak amplitude of 33.6, a settling time of 15.2 s, and a final value of 1.61. From the characteristic equation in (16), it can be observed that the system has nine poles in the left-half plane of the complex plane, indicating stability. However, dominant complex conjugate poles close to the imaginary axis are present. These poles exhibit oscillatory behavior, which is reflected in the oscillations and long settling time of the system. Such characteristics make the system highly susceptible to disturbances, which can lead to instability. In the context of this work, the root locus represents the roll-off phenomenon, where any changes in the model parameters can cause a sudden increase in tissue impedance, potentially disrupting the RFA procedure. The impending destabilization of the model highlights the need for a controller to modify the root locus and sustain a stable dynamic response of the RFA model.

settling time of 15.2 s and a final transfer function amplitude that differs from the expected value (1.61).

The Root Locus diagram, as depicted in Fig 9, illustrates the roll-off phenomenon, indicating that any changes in the model parameters can induce a sudden increase in tissue impedance, interrupting the RFA procedure. The impending destabilization of the model requires the presence of a controller to modify the Root Locus and ensure a stable dynamic response of the RFA model.

The evaluation of the system's Root Locus diagram in Fig 9, linked to the system's transient response, reveals the representation of the open-loop poles and zeros, noting the absence of a control process in the control parameters. As it is a qualitative method, it is possible to observe

**Table 2. Parameters applied to PSO.**

| . . . | Min. value | Max. value |
|---|---|---|
| S | 30 | . . . |
| N | 3 | . . . |
| $x$ | 0.01 | 10 |
| $\omega$ | 0.1 | 0.9 |
| $c_1$ | 2.05 | . . . |
| $c_2$ | 2.05 | . . . |
| $v$ | $\frac{v_{max}}{3}$ | $(x_{max} - x_{min}) \times 2$ |
| $MaxIter$ | . . . | 10 |
| $\alpha$ | 0.5 | . . . |
| $\beta$ | 0.6 | . . . |
| $\gamma$ | 0.8 | . . . |

that the dominant poles, close to the y-axis, have very short branches that, when varying any parameter, affect the root locus branch.

To make adjustments in this system, it is necessary to carefully evaluate the ideal positions for the inclusion of parameters, in order to meet stability and sensitivity criteria. This analysis provides an understanding of the best control strategy to be selected for the system, the need for simplifications in the extracted mathematical model, and the limitations of the model. This justifies the inclusion of a method for constructing a controller that generates a mathematical precision of adjustment and design made by using bioinspired systems techniques. This leads us to believe that controllers incorporated into ablation systems should be adjusted using techniques like those presented in this article. Table 5 presents the roots of the open-loop system.

## PSO results

We performed all simulations on a computer with a 2.2 GHz Intel® Core I5–5200 processor and 8 GB of RAM. We used MATLAB® / SIMULINK® 2021a. The computational cost will be presented in the following sections.

Table 2 presents the parameters used to obtain the best results. The performance of the PID controller was estimated using the PSO algorithm, and the results are shown in Fig 10 and Table 3. Points A, B and C in the step response graph indicate the rise time, overshoot and settling time, respectively. The step response illustrate the behavior of the PID controller described in (7), applied to the plant described in (16).

Additionally, the convergence curve of the PSO algorithm is displayed in Fig 11, demonstrating that the algorithm converged to the solution after 7 iterations, with a precision of $10^{-2}$.

By implementing the PID controller and introducing negative feedback, we confirmed that the system still satisfies the Routh-Hurwitz stability criterion, as described in Table 4. In contrast to the open-loop system, where the poles of the characteristic equation are considerably close to the zeros of the transfer function (TF), the introduction of the controller significantly influences the TF of the model. This influence can result in the poles experiencing negligible displacement towards their respective zeros with the variation of gain. This transformation can change a system that initially exhibited a long settling time into a critically damped system, as shown in the root locus diagram (see Fig 10).

With the introduction of the controller, the branches of the root locus were repositioned, notably in the interval between [x = -6 and -7] and [x = -2.5 and -3], optimizing their decay. These branches, when located on the real axis, contribute to adjusting the steady-state response

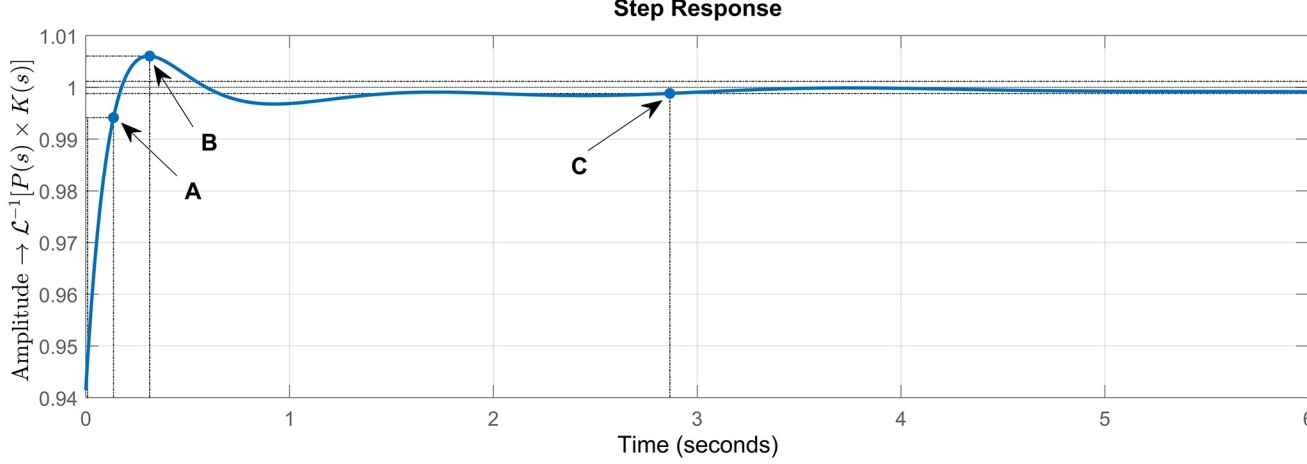

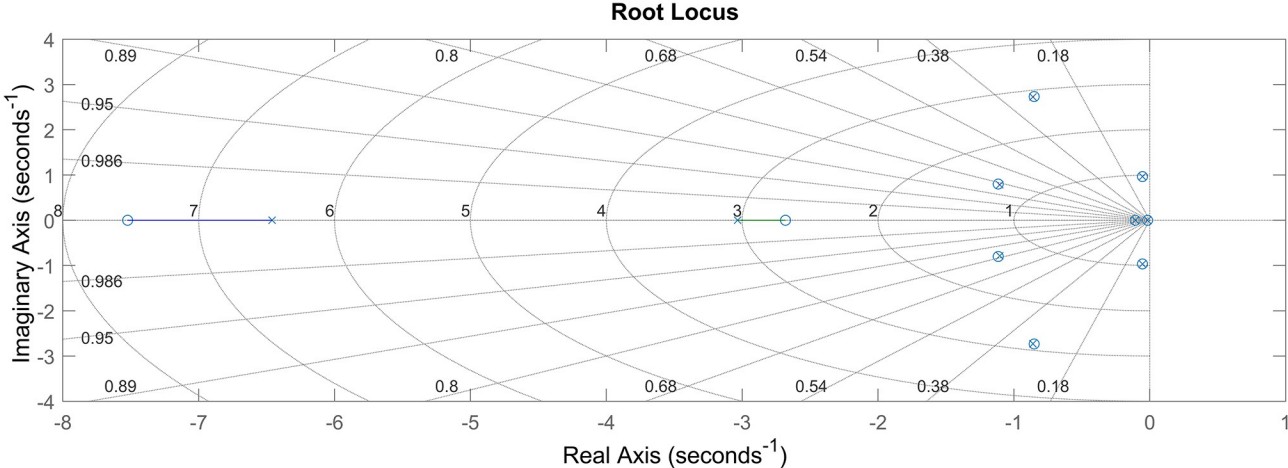

**Fig 10. Step response and root locus after PSO PID tuning.** The plots depict the response of the PSO algorithm with the parameters presented in Table 3. The step response illustrate the effect of the applied PID controller on P(s) (see (16)), where the input is voltage and the output is impedance, represented by the tuned TF K(s) (see (7)). Points A, B, and C highlight the instants where the rise time, peak amplitude, and settling time of the step response are measured, respectively. Additionally, the root locus after PID controller implementation is displayed. The poles of the characteristic equation are considerably close to the zeros of the TF. This indicates the significant influence of the controller on the TF of the model in such a way that the poles underwent a minimal displacement towards their respective zeros with the variation of the gain, transforming an underdamped system into a critically damped one.

**Table 3. Simulation PID gains and performance indexes.**

| $\cdots$ | $K_p$ | $K_i$ | $K_d$ |
|---|---|---|---|
| $\cdots$ | 5.0594 | 10.0 | 0.4959 |
| Peak amplitude | 1.007(B) | | |
| Overshoot | 0.605% at 0.314s (B) | | |
| Rise time (s) | 0.127 (A) | | |
| Settling time (s) | 2.87(C) | | |
| Elapsed time (s) | 832 | | |

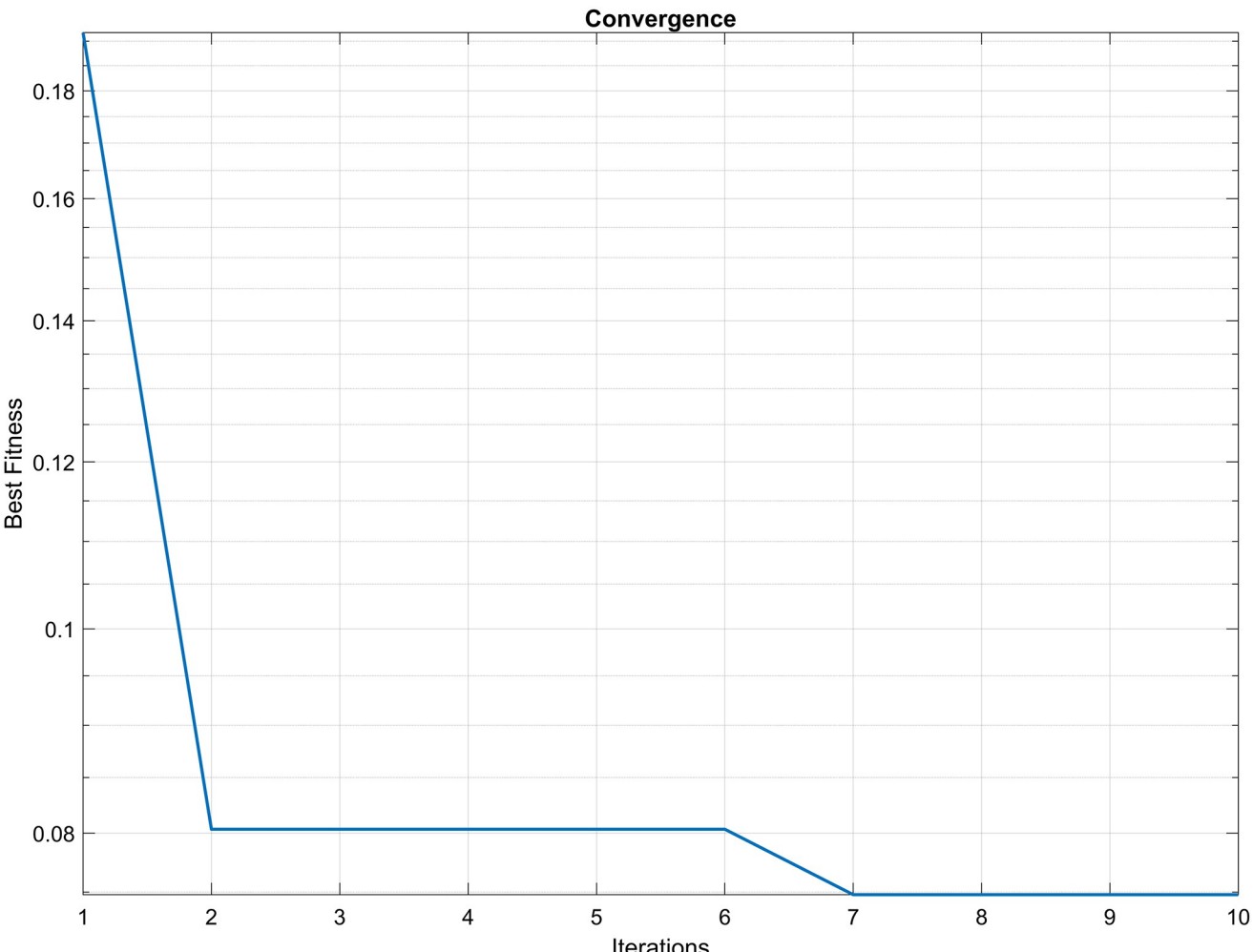

**Fig 11. PSO convergence curve.** The convergence curve of the PSO algorithm is displayed, demonstrating that the algorithm converged to the solution after 10 iterations, with a precision of $10^{-2}$.

**Table 4. Routh-Hurwitz stability criterion for the $P(s)+K(s)$ TF: Closed-loop roll-off system.**

| | | | | | | |
|---|---|---|---|---|---|---|
| $s^{10}$ | 17.08 | 1276.1 | 10389 | 15563 | 5874.3 | 8.12 |
| $s^9$ | 233.14 | 4412.4 | 15301 | 11168 | 592.91 | $\cdots$ |
| $s^8$ | 952.84 | 9268.2 | 14745 | 5830.9 | 8.12 | $\cdots$ |
| $s^7$ | 2144.7 | 11693 | 9741.5 | 590.93 | $\cdots$ | $\cdots$ |
| $s^6$ | 4073.3 | 10417 | 5568.4 | 8.12 | $\cdots$ | $\cdots$ |
| $s^5$ | 6208.2 | 6809.6 | 586.65 | $\cdots$ | $\cdots$ | $\cdots$ |
| $s^4$ | 5949.1 | 5183.4 | 8.12 | $\cdots$ | $\cdots$ | $\cdots$ |
| $s^3$ | 1400.4 | 578.17 | $\cdots$ | $\cdots$ | $\cdots$ | $\cdots$ |
| $s^2$ | 2727.3 | 8.12 | $\cdots$ | $\cdots$ | $\cdots$ | $\cdots$ |
| $s^1$ | 574 | $\cdots$ | $\cdots$ | $\cdots$ | $\cdots$ | $\cdots$ |
| $s^0$ | 8.12 | $\cdots$ | $\cdots$ | $\cdots$ | $\cdots$ | $\cdots$ |

**Table 5. Root locus for the open-loop and closed-loop system.**

| Open-loop system roots | | Closed-loop system roots | |
|---|---|---|---|
| **Zeros** | **Poles** | **Zeros** | **Poles** |
| -0.8509 + 2.7340i | -0.3521 + 3.1406i | -7.5226 + 0.0000i | -6.4591 + 0.0000i |
| -0.8509 − 2.7340i | -0.3521 − 3.1406i | -0.8509 + 2.7340i | -3.0329 + 0.0000i |
| -1.1152 + 0.8003i | -0.2576 + 1.6943i | -0.8509 − 2.7340i | -0.8595 + 2.7274i |
| -1.1152 − 0.8003i | -0.2576 − 1.6943i | -2.6809 + 0.0000i | -0.8595 − 2.7274i |
| -0.0538 + 0.9654i | -0.0979 + 0.9935i | -1.1152 + 0.8003i | -1.1037 + 0.7842i |
| -0.0538 − 0.9654i | -0.0979 − 0.9935i | -1.1152 − 0.8003i | -1.1037 − 0.7842i |
| -0.1067 + 0.0000i | -0.3052 + 0.3021i | -0.0538 + 0.9654i | -0.0540 + 0.9654i |
| -0.0162 + 0.0000i | -0.3052 − 0.3021i | -0.0538 − 0.9654i | -0.0540 − 0.9654i |
| · · · | -0.0934 + 0.0000i | -0.1067 + 0.0000i | -0.1067 + 0.0000i |
| · · · | · · · | -0.0162 + 0.0000i | -0.0162 + 0.0000i |

to reach the desired threshold. As for the other poles, especially the complex conjugate pairs (see Table 5), they were precisely adjusted with a precise branching locus, thanks to the presence of the PSO. This was done to ensure that performance indicators achieved the best possible response. The individual contributions of each pair, in terms of stability, were evenly distributed in the plane. By analyzing the angular contributions of each one, we can observe the balance that a PID controller adjusted by PSO can promote.

Consequently, we deduce that the RFA procedure moves away from the realm of imminent instability, allowing control of the roll-off by adjusting the input voltage while preserving the tissue impedance at a predetermined value.

In an additional set of 19 simulations using a swarm of 30 particles and 10 iterations, the proportional gain (Kp) exhibited a mean of 5.86, variance of 4.22, and a standard deviation of 2.05. The integral gain (Ki) showed values of 9.89, 0.048, and 0.22, respectively. Conversely, the derivative gain (Kd) displayed a mean of 0.57, variance of 0.021, and a standard deviation of 0.14, as we can see in Table 6. Furthermore, the analysis of variance (ANOVA) across the 19 experiments yielded a p-value $\ll 0.05$, suggesting non-normality among the means. Detailed results and statistical analyses can be found in the supplementary materials.

We also explored different parameter settings to optimize the PID controller design and assess the computational cost. Due to the random search nature of the PSO algorithm, each execution returns different controller parameters. We present the results of these simulations for the PID gains in Table 7. The execution time (in seconds) of the algorithm is closely related to the swarm population and the number of iterations, as well as the fixed TF of the plant. The simulation results show that the best trade-off between performance indices and execution time was obtained in Table 3.

In the last column of Table 7, we present a simulation with a significantly larger particle swarm, along with the algorithm being executed a hundred times. In this scenario, it can be inferred that the simulation time grew considerably and did not demonstrate an improved

**Table 6. Statistical analysis of controller gains.**

| Parameter | Mean | Variance | Standard Deviation |
|---|---|---|---|
| Kp | 5.86 | 4.22 | 2.05 |
| Ki | 9.89 | 0.048 | 0.22 |
| Kd | 0.57 | 0.021 | 0.14 |

**Table 7. Additional simulations.**

| $S$ | 10 | 20 | 20 | 50 | 30 | 100 |
|---|---|---|---|---|---|---|
| $MaxIter$ | 50 | 30 | 50 | 10 | 50 | 100 |
| $K_p$ | 3.7047 | 3.1824 | 10.0 | 10.0 | 10.0 | 4.0299 |
| $K_i$ | 10.0 | 9.9105 | 10.0 | 10.0 | 10.0 | 10.0 |
| $K_d$ | 0.4453 | 0.4477 | 0.8001 | 0.8005 | 0.8116 | 0.4698 |
| Elapsed time (s) | 1713 | 2018 | 3221 | 1704 | 4536 | 29737 |

In this table, we investigated other possible outcomes for the tuning of the PID controller parameters. From the perspective of PSO, the meta-heuristic returns different results for each simulation. Therefore, we varied the number of swarm individuals and the number of iterations to show the convergence level of the meta-heuristic and compare the computational cost for each initial condition application.

result compared to a swarm of 30 particles executing 10 iterations. The performance indices of this simulation for a step response were: 0.768% overshoot at 0.325 s, 0.138 s rise time, and 2.82 s settling time. More details on the simulation results can be found in the supplementary material.

## Discussion

The integration of control systems into RFA procedures for monitoring tissue impedance plays a crucial role in expanding the ablative volume and, consequently, in combating HCC. Enhancing the necrosis region beyond 3 cm has been the focus of numerous studies. However, the roll-off phenomenon, intrinsic to the carbonization of affected tissue in RFA, makes it crucial to regulate the power applied to the target region. One of the primary objectives in the RFA procedure is to avoid any disruption of the energy flow and heat propagation through the tissue.

In this study, we emphasized the application of a PID controller tuned using the PSO technique. Our focus is on managing tissue impedance to delay the occurrence of roll-off and indirectly regulate temperature. We highlighted the real-time management of the manipulated variable by exploring the best performance indicators for the control system. The use of SI based on the experiments described in this paper has yielded satisfactory results for the application of the PID controller. The analysis of various SI techniques resulted in varying levels of accuracy. In this work, we have addressed the combination of methodologies to obtain the best model fit over the experimental data.

The experimental data used in this study were also applied in the work of Da Fonseca *et al* [37]. Using the Box-Jenkins SI technique, [37] concluded that the dynamic response of the system led to abrupt variations in impedance and exhibited a long settling time, in agreement with the ex vivo experiments. This study demonstrated that the mathematical model used revealed that the transient response of the system could be improved by performing the infusion of 0.9% saline solution at 5˚C. On the other hand, the incorporation of a properly tuned PID controller yields enhanced results compared to the saline infusion technique, as presented by Da Fonseca *et al* [37]. In addition to the prolonged settling time, their system exhibited considerable overshoot.

In the current scientific landscape, only a few authors have presented studies that apply control methods targeting specific variables to enhance RFA procedures. In this context, Cheng *et al* [54] implemented temperature control in RFA interventions based on Fuzzy-PID control. Using voltage as the system input and temperature as the output, this study yielded significant results. By working with a third-order TF, the authors achieved commendable

performance indices for the dynamic model. Although this study did not describe the values of gains applied in the tuned PID.

To address the lack of systematic methods for calculating the parameters of the PID controller in control systems applied to RFA, we began by identifying experimental data using the ARX polynomial method, which we then transformed into the TF domain. This approach differs from that of Haemmerich and Webster [50], who identified the model using a discrete TF and the software ANA 2.52 (Freeware, Dept. of Control Engineering, Tech. Univ. Vienna/Austria), and determined the model parameters using the recursive least squares algorithm, as in the ARX method.

To model the RFA procedure more faithfully to the laboratory setting, where energy transfer and impedance variation occur in the continuous-time domain, we designed the PID controller from a continuous-time TF. This approach stands in contrast to that of [50], who used a discrete TF. It is noteworthy that while several theoretical studies on control systems applied to RFA exist, few offer systematic methods for calculating the PID parameters $K_p$, $K_i$, and $K_d$ [81]. This underscores the need for further research to develop more systematic approaches for designing PID controllers in RFA systems.

Alba-Martínez et al [81], recognizing that most commercial RFA equipment utilize a PI (Proportional Integral) controller, applied this model to control a dynamic system derived from a biological study simulated in COMSOL® Multiphysics software (COMSOL Inc., Burlington, MA, USA), and identified using MATLAB®. Similarly, Webster et al [50] employed a PI controller in their study. They, too, used the error between the current temperature of the active electrode and the set point temperature as the input signal of the controller. This error modulated the voltage applied between the active and dispersive electrodes, determining the output of the controller. The temperature measured at the active electrode served as the output signal of the system.

Using the PSO algorithm to determine the controller tunings resulted in significant outcomes. Alba-Martínez et al [81] utilized the geometric root locus technique to determine the PI controller gains, while Webster et al [50] chose the gains empirically to minimize overshoot and achieve a temporal behavior akin to in vivo experiments. The results found by [81] for the gains of the PI controller have values equal to $Kp = 4.78$ and $Ki = 3.39$. This study does not provide further details regarding performance indices. In the case of [50], the values are $Kp = 0.02$ and $Ki = 0.0064$, obtaining an overshoot of 11%, which was reached after 148 s.

Although the output variable of the dynamic system in our study differs, our simulations yielded significant results. Specifically, after conducting several simulations, the calculated controller gains shared a notable characteristic: the magnitude of the $K_d$ parameter was reduced when compared to the other gains (as shown in Table 7), bringing us closer to a PI controller.

This finding aligns with our expectations, as it suggests that the dynamic system model that describes the roll-off phenomenon based on tissue impedance exhibits clear similarities with previously studied models based on the active electrode temperature [24, 38]. The model presented in this paper offers the capability to simulate various conditions encountered during RFA interventions by utilizing a bioinspired search topology. It effectively avoids local minima in the objective function, which is strongly connected to the TF of the model.

From a computational simulation perspective, the controller's response suggests that there will be an effective shift in the roll-off curve or, conversely, the prevention of the phenomenon's occurrence. The next steps in this research will involve incorporating the control system into the SOFIA® equipment, along with the optimization algorithm, in the hope of obtaining a system that can be used in RFA of other tissues [37]. In this context, when the ex-vivo tests were conducted, we only have data regarding the ablation volume without the implementation

of the controller in the SOFIA device. The citations related to the coagulation volume were based on references that suggest that with the displacement of the roll-off curve, the procedure would not terminate, leading to a more extensive treatment.

The results of this study have the potential to significantly influence clinical guidelines for the treatment of hepatic tumors through RFA. The application of the PSO technique for tuning the PID controller may have a direct impact on the accuracy and effectiveness of RFA. This improvement may result in enhanced clinical outcomes for patients, including more effective tumor ablation, reduced associated complications, and possibly an increase in survival.

The optimized procedure that has emerged from this study comprises a series of well-defined steps intended to guide healthcare professionals in its implementation. Initially, it is crucial to ensure that the RFA equipment is properly calibrated and sterilized. Next, the administration of local anesthesia and correct patient positioning are performed. High-precision imaging techniques are employed for the precise localization of the tumor. The specific settings of the PID controller optimized by PSO are then applied, taking into consideration tissue impedance dynamics. The RFA procedure is initiated, with continuous monitoring of impedance and tissue temperature. Adjustments to the PID controller settings are made as necessary to maintain optimal ablation. Following ablation, an evaluation is conducted to determine the extent of tumor treatment and ensure patient stability.

By optimizing the PID controller, the procedure aims to minimize risks for both patients and the medical team during RFA treatments. The control precision provided by the PID optimized by PSO reduces the potential for complications. Additionally, there are considerations related to potential cost savings. However, specific data and cost estimates require further in-depth analysis. To facilitate the effective implementation of this method, it is important for medical professionals to acquire the necessary training and education.

Comparing the PID controller optimized by PSO approach with existing methods highlights its advantages. This approach offers precise control, minimizing the roll-off phenomenon and enhancing practical effectiveness compared to other techniques in use. To promote continuous improvement, it is crucial to establish a feedback mechanism that allows professionals to share their experiences and suggestions for procedure optimization. While real case studies or specific success stories of this approach may be limited in number, ongoing research and clinical trials can provide valuable practical evidence. These studies offer insights into the technique's effectiveness in terms of improving patient outcomes and optimizing the RFA procedure.

Through a thorough analysis of our model's dynamics, we can uncover valuable insights into the intricate interplay among tissue impedance, ablation temperature, and electrical factors in RFA. This model offers substantial benefits to both researchers and clinicians working in the realm of RFA, facilitating a comprehensive grasp of the fundamental dynamics that govern the ablation process.

## Limitations and future work

One limitation of the proposed model is related to the ex-vivo experiments, which did not take into account the effects of blood perfusion and the heating generated by metabolism, which are present in in vivo RFA scenarios and are also represented in the Pennes' equation. Additionally, the laboratory tests were performed on liver sections rather than the entire organ, which may not fully capture the behavior of electrothermal heating in a complete liver inside the body. Therefore, while the proposed model provides valuable insights into the dynamics of RFA, its applicability to in vivo scenarios may be limited and should be further validated in future studies.

As a prospective avenue for future research, the authors intend to explore the adoption of different objectives in the optimization of RFA procedures. One of these objectives could be to minimize the impact on the ablation zone in healthy tissue, with the aim of avoiding necrosis in the surrounding non-tumor tissue. This approach would prioritize the preservation of healthy liver tissue, which is essential for the patient's overall liver function.

Furthermore, an intriguing prospect for future research involves performing a multi-objective optimization analysis of RFA procedures, which, in addition to minimizing damage to healthy tissue, would also consider factors such as exposure time during the procedure. Multi-objective optimization would allow several parameters to be adjusted simultaneously, providing a more comprehensive assessment of the procedure's effectiveness.

In addition, we plan to investigate the effect of blood perfusion and metabolic heating on the RFA procedure and include these factors in our model. We will also explore the behavior of the RFA procedure in different organs and tissues, such as lung, pancreas, and kidney. Furthermore, we aim to develop a real-time control system that will allow us to monitor and adjust the RFA parameters during the procedure, based on feedback from the tissue impedance and temperature measurements. Finally, we will investigate the use of other types of controllers, such as adaptive and robust controllers, to improve the performance and robustness of the RFA procedure.

The authors believe that by exploring these future research directions, they can contribute to the advancement of RFA techniques, making them more precise, patient-centered and efficient. This approach has the potential to produce better outcomes for patients with liver tumors while minimizing the impact on healthy liver tissue, which is a critical consideration in the context of RFA.

## Conclusion

This paper presents a novel approach for controlling tissue impedance during RFA procedures based on porcine liver ex-vivo experiments. Our methodology combines three system identification techniques, yielding superior results compared to individual methods.

The implementation of a PID control technique effectively regulates the voltage at the active catheter tip relative to the dispersive one, thereby maintaining tissue impedance at a user-defined preset value to prevent roll-off. The PSO algorithm optimizes the controller gains, resulting in satisfactory performance indicators: a mere 0.605% overshoot, 0.314 seconds rise time, and 2.87 seconds settling time for a unit step input. This showcases the robustness of our PSO-tuned PID controller design in achieving precise and efficient control.

Unlike previous studies that primarily focus on modeling RFA based on tissue temperature or use trial and error techniques to determine the optimal voltage for roll-off delay, our work highlights the importance of controlling tissue impedance as a critical parameter for ensuring continuous and successful RFA.

Moreover, our results demonstrate that the PSO-tuned PID controller design is robust and adaptable to variations in liver tissue properties and environmental conditions. The statistical analysis of 19 simulations revealed PID gains: $Kp$ (mean: 5.86, variance: 4.22, standard deviation: 2.05), $Ki$ (mean: 9.89, variance: 0.048, standard deviation: 0.22), and $Kd$ (mean: 0.57, variance: 0.021, standard deviation: 0.14). The clinical implications of this study are significant and can lead to the development of more effective RFA procedures with reduced complications.

Future research should explore hardware implementations of the proposed PID controller and validate its performance in in-vivo animal models. Additionally, investigating the impact of controlled tissue impedance on the coagulation zone size and conducting clinical trials

will further enhance the understanding and applicability of our approach in real-world scenarios.

Overall, this study contributes to advancing the field of RFA by introducing a bioinspired solution for roll-off control, offering improved outcomes and expanding the possibilities for liver tumor treatment.

## Supporting information

**S1 Fig. This appendix presents the flowchart used in the development of the particle swarm optimization algorithm for determining the optimal PID controller gain values.** (TIF)

**S1 File. This file contains the spreadsheet with data from ex vivo tests used for the system identification process, resulting in the final 9th-order transfer function model.** (XLSX)

**S2 File. This file contains the validation data spreadsheet from the ex vivo test.** (XLSX)

**S3 File. This file contains the spreadsheet with data from the simulations, including statistical analysis and ANOVA calculations.** (XLSX)

**S4 File. This folder contains the implemented codes for system identification using the estimation and validation data sheets.** Additionally, the folder includes the necessary files to reproduce the execution of the particle swarm optimization algorithm and a "read me" file that provides instructions for running the codes in question. (ZIP)

## Acknowledgments

The authors would like to express their gratitude to God for endowing each researcher with the skills necessary for the advancement of scientific research aimed at enhancing ablation procedures and improving quality of life. We also extend our thanks to the scientific journal PLOS ONE for accepting and disseminating this work. Additionally, we are grateful to everyone who contributed in any way to the development of this study.

## Author Contributions

**Conceptualization:** Rafael Mendes Faria, Suélia de Siqueira Rodrigues Fleury Rosa, Ana Karoline Almeida da Silva.

**Data curation:** Rafael Mendes Faria, Suélia de Siqueira Rodrigues Fleury Rosa, Gustavo Adolfo Marcelino de Almeida Nunes, Klériston Silva Santos, Rafael Pissinati de Souza, Ana Karoline Almeida da Silva, Sylvia de Sousa Faria.

**Formal analysis:** Rafael Mendes Faria, Ana Karoline Almeida da Silva.

**Methodology:** Rafael Mendes Faria, Suélia de Siqueira Rodrigues Fleury Rosa, Gustavo Adolfo Marcelino de Almeida Nunes, Klériston Silva Santos, Rafael Pissinati de Souza, Ana Karoline Almeida da Silva, Mario Fabrício Rosa, Sylvia de Sousa Faria.

**Project administration:** Suélia de Siqueira Rodrigues Fleury Rosa.

**Software:** Rafael Mendes Faria.

**Supervision:** Suélia de Siqueira Rodrigues Fleury Rosa.

**Writing – original draft:** Rafael Mendes Faria, Ana Karoline Almeida da Silva.

**Writing – review & editing:** Suélia de Siqueira Rodrigues Fleury Rosa, Gustavo Adolfo Marcelino de Almeida Nunes, Rafael Pissinati de Souza, Angie Daniela Ibarra Benavides, Angélica Kathariny de Oliveira Alves, Mario Fabrício Rosa, Antônio Aureliano de Anicêsio Cardoso, Sylvia de Sousa Faria, Enrique Berjano, Adson Ferreira da Rocha, Ícaro dos Santos, Ana González-Suárez.

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
