## [Decision Letter · Decision Letter 0]

11 Sep 2023

PONE-D-23-20054Bio-inspired solution for roll-off control in radiofrequency ablation of liver tumors: a particle swarm optimization approach for PID controller tuningPLOS ONE

Dear Dr. Faria,

Thank you for submitting your manuscript to PLOS ONE. After careful consideration, we feel that it has merit but does not fully meet PLOS ONE’s publication criteria as it currently stands. Therefore, we invite you to submit a revised version of the manuscript that addresses the points raised during the review process.

We look forward to receiving your revised manuscript.

Kind regards,

Salim Heddam

Academic Editor

PLOS ONE

2. Please amend the manuscript submission data (via Edit Submission) to include author Enrique Berjano.

3. Please amend your authorship list in your manuscript file to include author Henrique Berjano.

4. We are unable to open your Supporting Information file [Supporting information.rar]. Please kindly revise as necessary and re-upload.

Additional Editor Comments:

Reviewer 1:This paper deals about the optimization of a PID controller used for liver tumors radio-frequency ablation. After performing ex-vivo experiments, the optimization model is set up by considering the roll-off - i. e., the impedance - as the objective function, and the PID parameters as the input function. The optimization predictive model is built up with some advanced regression analysis. After a throughout analysis to verify that the algorithm performs well, the authors conclude that this optimization might be very helpful for whom wants to improve the PID procedure accounting for the roll-off phenomenon.

The reviewer thinks that this is an interesting contribution that would be really helpful for people that practice radiofrequency ablation, since they have some hints to avoid roll-off. It is then suggested to consider the present paper for publication after the authors address the following points.

- In the title, the authors use the word "bio-inspired" to characterize what they do here. Are they making references to the fact that a bio-inspired optimization algorithm, say particle swarm optimization, is used? If so, please change the word "bio-inspired" with something like "particle swarm optimization", "optimum search", or similar. This would avoid any misunderstandings for the readers

- Among various optimization techniques, why didn't the authors use other methods like genetic algorithm (doi.org/10.1007/s11042-020-10139-6), bee colony algorithm (doi.org/10.1007/s10462-012-9328-0), and so on? These methods allow to have shorter computational times

- When writing the paper, it is suggested to avoid too many new lines since they make the text less fluent

- When introducing the experimental protocol, are the authors sure that the experiments are design in order to avoid any heat losses to the environment? These losses might influence the ablated zone

- In the model structure section, the authors report that the equipment used doesn't allow to measure at every constant time step. Is it reasonable to perform some linear interpolation? Please report some qualitative information about these time steps at which the equipment is able to perform measurements

- The authors claim that they use the ARX method to have some regressions between input and output (Eq. 1). Why didn't they use standard regressions or other AI techniques like for instance artificial neural networks?

- The reviewer thinks that the paper introduction is too much focused about roll-off phenomenon, even if in this field - tumor thermal ablation - other problems might arise and require several attention from the scientific community. These problems might be for instance tissue shinkrage (doi.org/10.1371/journal.pone.0210667) or underdesired healthy tissue damage (doi.org/10.3390/ma13010136), that might be treated with approaches based on customized procedures (doi.org/10.1016/j.cmpb.2020.105887) or different antennas to be used (doi.org/10.1080/10407782.2020.1835083). In order to underline the fact that optimizing radio-frequency ablation procedures - a widely used technique - still remains an open problem, the authors should consider all these aspects when writing their paper introduction.

- When validating the regression model, it is suggested to use some graphs to show that the coefficient of determination (Eq. 6) is not that high; a graph that report experimental outcomes (x-axis) and regressed model (y-axis) with the bisector could be an idea

- When describing the particle swarm optimization algorithm, even if the text is quite clear, it is suggested to show a figure/table that clearly presents input variables (to be changed), constraints (to be not modified) and objective function; the authors could also integrate this within Fig. 1

- When the optimum solution is obtained, it is suggested to report a comparison between the optimum solution and a reference solution, to appreciate how important is to run optimization for this specific case

- In the manuscript, with particular references to abstract and discussion sections, it is suggested to provide more practical outcomes from the present works, like for instance more details about the optimum procedure that has been found. This would be really helpful in medical practice

- As a potential future work, the authors could also think about considering a different objective function, like for instance the ablation zone with no necrosis for the healthy tissue. Another idea could be to run a multi-objective optimization analysis considering also aspects like exposure time

Reviewer 2:Dear Sir,

The paper deserves the publication after carefully satisfying the following points:

1. The abstract is very lengthy. Rewrite it in a concise manner and mention in it the obtained findings and show some statistical numbers.

2. The manuscript needs a comprehensive English language proofreading, there are many linguistic mistakes.

3. The introduction does not provide sufficient background information for readers not in the immediate field to understand the problem/hypotheses.

4. The literature survey is not balanced, important studies are not cited in this work, the following papers are very close to the topic of the paper and must be cited,

• doi:10.1177/1729881420981524.

• https://doi.org/10.1049/tje2.12009.

• https://doi.org/10.1016/B978-0-12-820276-0.00026-1.

• doi.org/10.1177/0020294020917171.

• https://doi.org/10.1155/2020/3067024.

5. I think the motivations for this study need to be made clearer.

6. The study objectives must be clearly defined.

7. no comparisons found with recent works, these are necessary to verify the effectiveness of the proposed designs.

8. The results are not clearly explained and must be presented in an appropriate format.

9. Stability analysis of the closed-loop system in the presence of the proposed ESO is not available, it must be addressed.

10. The findings are not properly described in the context of the published literature.

11. No significant limitations are discussed. It may be worthwhile to mention the tradeoffs involved in your work.

Reviewer 3:1.Results shown can be shown in atabular form.

2.Some recent works should be referred in literature and in work.

3.Graphical analysis should be presented .

4.Conclusion should include quantitative results.

Reviewers' comments:

Reviewer's Responses to Questions

**Comments to the Author**

1. Is the manuscript technically sound, and do the data support the conclusions?

Reviewer #1: Yes

Reviewer #2: Partly

Reviewer #3: Yes

2. Has the statistical analysis been performed appropriately and rigorously? 

Reviewer #1: N/A

Reviewer #2: N/A

Reviewer #3: Yes

3. Have the authors made all data underlying the findings in their manuscript fully available?

Reviewer #1: Yes

Reviewer #2: Yes

Reviewer #3: No

4. Is the manuscript presented in an intelligible fashion and written in standard English?

Reviewer #1: Yes

Reviewer #2: No

Reviewer #3: Yes

5. Review Comments to the Author

Reviewer #1: This paper deals about the optimization of a PID controller used for liver tumors radio-frequency ablation. After performing ex-vivo experiments, the optimization model is set up by considering the roll-off - i. e., the impedance - as the objective function, and the PID parameters as the input function. The optimization predictive model is built up with some advanced regression analysis. After a throughout analysis to verify that the algorithm performs well, the authors conclude that this optimization might be very helpful for whom wants to improve the PID procedure accounting for the roll-off phenomenon.

The reviewer thinks that this is an interesting contribution that would be really helpful for people that practice radiofrequency ablation, since they have some hints to avoid roll-off. It is then suggested to consider the present paper for publication after the authors address the following points.

- In the title, the authors use the word "bio-inspired" to characterize what they do here. Are they making references to the fact that a bio-inspired optimization algorithm, say particle swarm optimization, is used? If so, please change the word "bio-inspired" with something like "particle swarm optimization", "optimum search", or similar. This would avoid any misunderstandings for the readers

- Among various optimization techniques, why didn't the authors use other methods like genetic algorithm (doi.org/10.1007/s11042-020-10139-6), bee colony algorithm (doi.org/10.1007/s10462-012-9328-0), and so on? These methods allow to have shorter computational times

- When writing the paper, it is suggested to avoid too many new lines since they make the text less fluent

- When introducing the experimental protocol, are the authors sure that the experiments are design in order to avoid any heat losses to the environment? These losses might influence the ablated zone

- In the model structure section, the authors report that the equipment used doesn't allow to measure at every constant time step. Is it reasonable to perform some linear interpolation? Please report some qualitative information about these time steps at which the equipment is able to perform measurements

- The authors claim that they use the ARX method to have some regressions between input and output (Eq. 1). Why didn't they use standard regressions or other AI techniques like for instance artificial neural networks?

- The reviewer thinks that the paper introduction is too much focused about roll-off phenomenon, even if in this field - tumor thermal ablation - other problems might arise and require several attention from the scientific community. These problems might be for instance tissue shinkrage (doi.org/10.1371/journal.pone.0210667) or underdesired healthy tissue damage (doi.org/10.3390/ma13010136), that might be treated with approaches based on customized procedures (doi.org/10.1016/j.cmpb.2020.105887) or different antennas to be used (doi.org/10.1080/10407782.2020.1835083). In order to underline the fact that optimizing radio-frequency ablation procedures - a widely used technique - still remains an open problem, the authors should consider all these aspects when writing their paper introduction.

- When validating the regression model, it is suggested to use some graphs to show that the coefficient of determination (Eq. 6) is not that high; a graph that report experimental outcomes (x-axis) and regressed model (y-axis) with the bisector could be an idea

- When describing the particle swarm optimization algorithm, even if the text is quite clear, it is suggested to show a figure/table that clearly presents input variables (to be changed), constraints (to be not modified) and objective function; the authors could also integrate this within Fig. 1

- When the optimum solution is obtained, it is suggested to report a comparison between the optimum solution and a reference solution, to appreciate how important is to run optimization for this specific case

- In the manuscript, with particular references to abstract and discussion sections, it is suggested to provide more practical outcomes from the present works, like for instance more details about the optimum procedure that has been found. This would be really helpful in medical practice

- As a potential future work, the authors could also think about considering a different objective function, like for instance the ablation zone with no necrosis for the healthy tissue. Another idea could be to run a multi-objective optimization analysis considering also aspects like exposure time

Reviewer #2: Dear Sir,

The paper deserves the publication after carefully satisfying the following points:

1. The abstract is very lengthy. Rewrite it in a concise manner and mention in it the obtained findings and show some statistical numbers.

2. The manuscript needs a comprehensive English language proofreading, there are many linguistic mistakes.

3. The introduction does not provide sufficient background information for readers not in the immediate field to understand the problem/hypotheses.

4. The literature survey is not balanced, important studies are not cited in this work, the following papers are very close to the topic of the paper and must be cited,

• doi:10.1177/1729881420981524.

• https://doi.org/10.1049/tje2.12009.

• https://doi.org/10.1016/B978-0-12-820276-0.00026-1.

• doi.org/10.1177/0020294020917171.

• https://doi.org/10.1155/2020/3067024.

5. I think the motivations for this study need to be made clearer.

6. The study objectives must be clearly defined.

7. no comparisons found with recent works, these are necessary to verify the effectiveness of the proposed designs.

8. The results are not clearly explained and must be presented in an appropriate format.

9. Stability analysis of the closed-loop system in the presence of the proposed ESO is not available, it must be addressed.

10. The findings are not properly described in the context of the published literature.

11. No significant limitations are discussed. It may be worthwhile to mention the tradeoffs involved in your work.

regards

Reviewer #3: 1.Results shown can be shown in atabular form.

2.Some recent works should be referred in literature and in work.

3.Graphical analysis should be presented .

4.Conclusion should include quantitative results.

6. PLOS authors have the option to publish the peer review history of their article (what does this mean?). If published, this will include your full peer review and any attached files.

Reviewer #1: No

Reviewer #2: No

Reviewer #3: No

---

## [Author Response · Author response to Decision Letter 0]

16 Nov 2023

REBUTTAL LETTER

Dear Reviewers,

We extend our sincere gratitude for your thoughtful reviews and valuable insights into our manuscript. Your constructive feedback has been instrumental in refining our work. We have diligently endeavored to address each of your recommendations to the best of our ability, striving to enhance the clarity, rigor, and overall quality of the paper.

Your expertise and guidance have been immensely valuable in shaping the manuscript into a more robust and coherent piece of research. We appreciate the time and effort you dedicated to the evaluation of our work.

Should you have any further suggestions or if there are additional aspects that you deem pertinent for improvement, we are more than willing to accommodate them. Your feedback is crucial to the advancement of our research, and we are committed to ensuring that the final version of the manuscript meets the highest standards.

Review 01:

1) In the title, the authors use the word "bio-inspired" to characterize what they do here. Are they making references to the fact that a bio-inspired optimization algorithm, say particle swarm optimization, is used? If so, please change the word "bio-inspired" with something like "particle swarm optimization", "optimum search", or similar. This would avoid any misunderstandings for the readers.

Answer: Dear reviewer, thank you for this and all other suggestions regarding the manuscript submitted to the PLoS ONE journal. We will implement all recommendations to ensure our article fully meets the publication criteria set by PLoS ONE. Accordingly, we present the new title: "Particle Swarm Optimization Solution for Roll-off Control in Radiofrequency Ablation of Liver Tumors: Optimal Search for PID Controller Tuning."

2) Among various optimization techniques, why didn't the authors use other methods like genetic algorithm (doi.org/10.1007/s11042-020-10139-6), bee colony algorithm (doi.org/10.1007/s10462-012-9328-0), and so on? These methods allow to have shorter computational times.

Answer: First and foremost, I would like to express gratitude for your inquiry regarding the choice of optimization technique and for the suggestions and corrections proposed. It's important to acknowledge awareness of other optimization algorithms, such as the genetic algorithm (https://doi.org/10.1007/s11042-020-10139-6), the bee colony algorithm (https://doi.org/10.1007/s10462-012-9328-0), among others.

The decision to use Particle Swarm Optimization (PSO) was based on several considerations. Firstly, PSO demonstrated robust performance in estimating controller gains, resulting in performance indices such as overshoot, rise time, and steady-state error that met our quality criteria.

Furthermore, PSO was chosen due to its successful integration in other applications and research conducted by our research group. Specifically, we are implementing the PSO algorithm in the hardware of the SOFIA equipment, a crucial device for our studies. An example of work with the SOFIA equipment can be found in our previous publication (https://doi.org/10.1109/TBME.2018.2873141).

It's essential to highlight that the current study under review is based on data obtained from the settings and results of this previous study. Therefore, the choice of PSO was motivated both by its reliable performance and our familiarity with its application in our research line.

I hope this information clarifies the selection of PSO as the optimization method in our article. Once again, we appreciate your interest and contribution to improving our work.

3) When writing the paper, it is suggested to avoid too many new lines since they make the text less fluente.

Answer: Dear reviewer. We sincerely appreciate your valuable suggestions and the thorough review of our paper. We have carefully considered all your comments, especially regarding line breaks, and have taken the necessary steps to ensure the smooth flow of the text. We highly value your contribution to the quality of our work and are confident that the implemented changes will significantly enhance the clarity and readability of the article. Once again, we thank you for your time and effort devoted to reviewing our manuscript.

4) When introducing the experimental protocol, are the authors sure that the experiments are design in order to avoid any heat losses to the environment? These losses might influence the ablated zone.

Answer: Thank you for the pertinent observation regarding potential heat losses during our ex-vivo experiments. In our laboratory experiments, we implemented various measures to minimize heat dissipation to the surroundings. We carefully insulated the porcine liver samples using materials with low thermal conductivity, such as acrylic. This material was strategically positioned, limiting the samples' contact to a window of only 6 cm per side, which connected to the grounding surface. Additionally, we employed effective thermal insulation on the tray where the experiments were conducted. Furthermore, we maintained the laboratory temperature at approximately 22°C, relying on air conditioning systems to create an environment that replicated the same conditions for all conducted tests.

5) In the model structure section, the authors report that the equipment used doesn't allow to measure at every constant time step. Is it reasonable to perform some linear interpolation? Please report some qualitative information about these time steps at which the equipment is able to perform measurements.

Answer: The SOFIA equipment used in our ex-vivo experiments did not automatically export data at constant time intervals. Consequently, the collection of measurements, including impedance, voltage, current, and power, was done manually, leading to variations in the time intervals between each acquisition.

To address this variability in measurement time intervals, we utilized the "resample" command available in the MATLAB software. This command was employed to resample the non-uniform data and adjust it to a fixed rate using interpolation techniques. In the manuscript, we provide a more detailed and comprehensive description of how this tool was used to process the data and ensure the consistency of measurements over time.

We hope this information has clarified the procedure adopted to deal with variations in measurement time intervals. We appreciate you bringing this issue to our attention and trust that these explanations have been helpful.

6) The authors claim that they use the ARX method to have some regressions between input and output (Eq. 1). Why didn't they use standard regressions or other AI techniques like for instance artificial neural networks?

Answer: Dear reviewer, we appreciate your careful attention to our paper and the pertinent observation regarding the system identification method employed. We chose the AutoRegressive with eXogenous input (ARX) method to establish regressions between input and output (Eq. 1) based on considerations of simplicity and effectiveness for our specific objectives. The selection of the ARX method was motivated by its simplicity, aiding in result interpretation and understanding of the system's behavior. Additionally, ARX offers a transparent approach to modeling dynamic systems, which was crucial for our analysis.

As for not employing other techniques, such as artificial neural networks, it is a valid consideration. However, our primary focus was on simplicity and ease of incorporating the model into the SOFIA equipment, aiming for an efficient solution that is easily implementable in clinical practice. We believe that the chosen approach, combining ARX with PSO, provides a balanced solution for the specific objectives of our study.

7) The reviewer thinks that the paper introduction is too much focused about roll-off phenomenon, even if in this field - tumor thermal ablation - other problems might arise and require several attention from the scientific community. These problems might be for instance tissue shinkrage (doi.org/10.1371/journal.pone.0210667) or underdesired healthy tissue damage (doi.org/10.3390/ma13010136), that might be treated with approaches based on customized procedures (doi.org/10.1016/j.cmpb.2020.105887) or different antennas to be used (doi.org/10.1080/10407782.2020.1835083). In order to underline the fact that optimizing radio-frequency ablation procedures - a widely used technique - still remains an open problem, the authors should consider all these aspects when writing their paper introduction.

Answer: We appreciate the valuable guidance, which will undoubtedly enrich the context and relevance of our work in the scientific literature. We acknowledge the importance of broadening the focus beyond the roll-off phenomenon and addressing other critical challenges in the field of thermal tumor ablation. Apart from roll-off, there are crucial issues such as tissue shrinkage (doi.org/10.1371/journal.pone.0210667) and undesired damage to healthy tissues (doi.org/10.3390/ma13010136) that demand attention from the scientific community. Considering the complexity of these problems, we will emphasize in our introduction the need for a comprehensive approach to optimize radiofrequency ablation procedures. This will include exploring customized procedures (doi.org/10.1016/j.cmpb.2020.105887) and the potential use of different antennas (doi.org/10.1080/10407782.2020.1835083) to address these multifaceted issues. This broader approach will reflect the complexity of the clinical scenario and the ongoing quest for effective solutions in thermal ablation, thereby contributing to a more holistic understanding of the field.

8) When validating the regression model, it is suggested to use some graphs to show that the coefficient of determination (Eq. 6) is not that high; a graph that report experimental outcomes (x-axis) and regressed model (y-axis) with the bisector could be an idea.

Answer: We appreciate the valuable suggestion. The requested graph has already been incorporated into Figure 5 of the manuscript. In this representation, it is possible to visualize the relationship between the experimental results and the fitted model. It is essential to highlight that, due to the high coefficient of determination, the measured data and those obtained in the regression are practically overlapped, indicating good agreement. To ensure full transparency, all calculations used in this analysis are thoroughly described in the ARX code, available as part of the supplementary material..

9) When describing the particle swarm optimization algorithm, even if the text is quite clear, it is suggested to show a figure/table that clearly presents input variables (to be changed), constraints (to be not modified) and objective function; the authors could also integrate this within Fig. 1.

Answer: We greatly appreciate the valuable suggestion for improving our article. In response to your recommendation, we have added an additional block to Figure 1, and its corresponding expansion is now highlighted in Figure 2, clearly presenting all input variables (to be changed), constraints (not to be modified), and the objective function. We hope that these changes significantly enhance the understanding of the PSO algorithm, providing a more comprehensive and elucidative visual representation..

10) When the optimum solution is obtained, it is suggested to report a comparison between the optimum solution and a reference solution, to appreciate how important is to run optimization for this specific case.

Answer: We appreciate the reviewer's suggestion, and we have included a detailed comparison between the obtained optimum solution and a reference solution in the manuscript. This comparison provides insights into the significance of running the optimization for this specific case, as suggested. The relevant information and analysis can be found in discussion of the revised manuscript.

11) In the manuscript, with particular references to abstract and discussion sections, it is suggested to provide more practical outcomes from the present works, like for instance more details about the optimum procedure that has been found. This would be really helpful in medical practice.

Answer: Dear Reviewers, we wish to express our sincere appreciation for the invaluable feedback and constructive comments you provided in your evaluation of the manuscript. Your wealth of experience and insightful guidance are of immense importance to us, and we hold your expertise in the highest regard. We extend our gratitude for the time and meticulous attention you dedicated to proofreading; we deeply acknowledge the significance of your thorough review.

Regarding your request, we wholeheartedly understand the importance of presenting more practical results, especially within the summary and discussion sections of our article. Your suggestion to furnish additional details on the optimal procedure revealed by our research aligns seamlessly with our commitment to enhancing the clinical applicability of our work.

In response, we have implemented the following adjustments to the Abstract:

• Rewritten the Abstract to enhance clarity and conciseness.

• Emphasized the potential to reduce incomplete tumor ablation, improve patient outcomes, and diminish tumor recurrence rates.

• Clarified the objective statement to facilitate better comprehension.

In response, we have implemented the following adjustments to the Discussion:

In the discussion, adjustments were made to emphasize the clinical significance of the study, provide a detailed description of the optimized procedure, highlight safety and cost considerations, underscore the need for training and comprehensive adoption, compare the new approach to existing methods, and emphasize the need for future research. The discussion concluded with comments on the study's potential and next steps.

12) As a potential future work, the authors could also think about considering a different objective function, like for instance the ablation zone with no necrosis for the healthy tissue. Another idea could be to run a multi-objective optimization analysis considering also aspects like exposure time.

Answer: We want to express our sincere gratitude to the reviewers for their invaluable insights and innovative suggestions. We are committed to incorporating objective function analysis, especially focused on protecting healthy tissue and eliminating satellite tumors in our future work. Your insights have significantly enriched our research and opened new possibilities to enhance the outcomes of our ablation procedures. Once again, we thank you for your invaluable contribution.

We've added 3 more paragraphs to the topic to meet the reviewers' demands. 

Review 02:

1. The abstract is very lengthy. Rewrite it in a concise manner and mention in it the obtained findings and show some statistical numbers.

Answer: Thank you for your valuable feedback. We have revised the abstract to enhance conciseness while highlighting key findings and incorporating statistical data. The updated abstract now provides a succinct overview of the study, including significant quantitative results, contributing to a more focused and informative presentation. 

2. The manuscript needs a comprehensive English language proofreading, there are many linguistic mistakes.

Answer: Thank you for the observation. We have enlisted the services of the university's English correction service to enhance the linguistic quality of the manuscript. However, we acknowledge the importance of linguistic accuracy, and if necessary, we are open to considering the hiring of PLOS ONE's revision services to ensure excellence in the article's presentation. We are committed to ensuring that the manuscript meets the required quality standards.

3. The introduction does not provide sufficient background information for readers not in the immediate field to understand the problem/hypotheses.

Answer: We sincerely thank you for your valuable observations and comments. We have taken your suggestions on board and, based on them, carried out a comprehensive revision of the article's introduction. We have made significant changes, including rewriting paragraphs and adding more references to provide a more solid foundation for reade

---

## [Decision Letter · Decision Letter 1]

8 Jan 2024

PONE-D-23-20054R1Particle swarm optimization solution for roll-off control in radiofrequency ablation of liver tumors: Optimal search for PID controller tuningPLOS ONE

Dear Dr. Faria,

Thank you for submitting your manuscript to PLOS ONE. After careful consideration, we feel that it has merit but does not fully meet PLOS ONE’s publication criteria as it currently stands. Therefore, we invite you to submit a revised version of the manuscript that addresses the points raised during the review process.

We look forward to receiving your revised manuscript.

Kind regards,

Salim Heddam

Academic Editor

PLOS ONE

Additional Editor Comments:

Reviewer 1#:

The reviewer thinks that the present paper can be accepted as it is in the revised form

Reviewer 4#:

Please improve figures quality.

Please add proper flowchart.

Please add block diagram.

Please justify gap and its addressing by your paper in literature.

Reviewer 5#:

it is a detailed theoretical and experimental study and the mostly the reviewers' comments are responded to; however, still, the following comments should be addressed:

-Careful attention should be given to correcting writing mistakes.

-Although the introduction and related work have been presented, the differences of the study should have been briefly highlighted in related work.

-I recommend referencing the following publication for PSO and PID techniques: https://doi.org/10.1007/978-3-031-26876-2_85. Additionally, more recent references from recent years should be included.

-In Fig 4, the peak point of the voltage, the stabil voltage, and applied voltage have not been explained in detail. The Voltage stabil time and later peak voltage should be explined.

-The meaning of the map and numbers in the root locus should be explained more thoroughly.

Overall ,except Figures 1 and 2, there is lack of detail in explanation for all the figures.

Reviewer 6#:

The paper presents a novel approach for controlling tissue impedance during RFA 710 procedures based on porcine liver ex-vivo experiments. The methodology combines 711 three system identification techniques, yielding superior results compared to individual 712 methods.

The paper is well written and organized in all sections, the results were well presented, likewise, the results from the analyses were discussed elaborately in an informative manner and were convincing, as well as the conclusion. The research novelty is somewhat minimal but scholarly convincing, as the discussions and analyses are based on existing technology but with little modification.

I hereby recommend the acceptance of this manuscript in its current form.

REMARKS

1. The title of the manuscript is appropriate.

2. The manuscript is technically sound and of High scientific quality.

3. The manuscript is free from errors and the grammar is satisfactory.

4. The tables and figures are clear.

5. The subject matter is presented comprehensively.

6. The references provided are all applicable but not sufficient because there are some parts in the paper that need to be cited appropriately.

Reviewers' comments:

Reviewer's Responses to Questions

**Comments to the Author**

1. If the authors have adequately addressed your comments raised in a previous round of review and you feel that this manuscript is now acceptable for publication, you may indicate that here to bypass the “Comments to the Author” section, enter your conflict of interest statement in the “Confidential to Editor” section, and submit your "Accept" recommendation.

Reviewer #1: All comments have been addressed

Reviewer #4: All comments have been addressed

Reviewer #5: All comments have been addressed

Reviewer #6: All comments have been addressed

2. Is the manuscript technically sound, and do the data support the conclusions?

Reviewer #1: Yes

Reviewer #4: Yes

Reviewer #5: Partly

Reviewer #6: Yes

3. Has the statistical analysis been performed appropriately and rigorously? 

Reviewer #1: N/A

Reviewer #4: N/A

Reviewer #5: Yes

Reviewer #6: Yes

4. Have the authors made all data underlying the findings in their manuscript fully available?

Reviewer #1: Yes

Reviewer #4: Yes

Reviewer #5: Yes

Reviewer #6: Yes

5. Is the manuscript presented in an intelligible fashion and written in standard English?

Reviewer #1: Yes

Reviewer #4: Yes

Reviewer #5: Yes

Reviewer #6: Yes

6. Review Comments to the Author

Reviewer #1: (No Response)

Reviewer #4: Please improve figures quality.

Please add proper flowchart.

Please add block diagram.

Please justify gap and its addressing by your paper in literature.

Reviewer #5: it is a detailed theoretical and experimental study and the mostly the reviewers' comments are responded to; however, still, the following comments should be addressed:

-Careful attention should be given to correcting writing mistakes.

-Although the introduction and related work have been presented, the differences of the study should have been briefly highlighted in related work.

-I recommend referencing the following publication for PSO and PID techniques: https://doi.org/10.1007/978-3-031-26876-2_85. Additionally, more recent references from recent years should be included.

-In Fig 4, the peak point of the voltage, the stabil voltage, and applied voltage have not been explained in detail. The Voltage stabil time and later peak voltage should be explined.

-The meaning of the map and numbers in the root locus should be explained more thoroughly.

Overall ,except Figures 1 and 2, there is lack of detail in explanation for all the figures.

Reviewer #6: The paper presents a novel approach for controlling tissue impedance during RFA 710 procedures based on porcine liver ex-vivo experiments. The methodology combines 711 three system identification techniques, yielding superior results compared to individual 712 methods.

The paper is well written and organized in all sections, the results were well presented, likewise, the results from the analyses were discussed elaborately in an informative manner and were convincing, as well as the conclusion. The research novelty is somewhat minimal but scholarly convincing, as the discussions and analyses are based on existing technology but with little modification.

I hereby recommend the acceptance of this manuscript in its current form.

REMARKS

1. The title of the manuscript is appropriate.

2. The manuscript is technically sound and of High scientific quality.

3. The manuscript is free from errors and the grammar is satisfactory.

4. The tables and figures are clear.

5. The subject matter is presented comprehensively.

6. The references provided are all applicable but not sufficient because there are some parts in the paper that need to be cited appropriately.

7. PLOS authors have the option to publish the peer review history of their article (what does this mean?). If published, this will include your full peer review and any attached files.

Reviewer #1: No

Reviewer #4: No

Reviewer #5: No

Reviewer #6: **Yes: **Dr. Peter Anuoluwapo Gbadega

---

## [Author Response · Author response to Decision Letter 1]

22 Feb 2024

REBUTTAL LETTER

Dear Reviewers,

We extend our sincere gratitude for your thoughtful reviews and valuable insights into our manuscript. Your constructive feedback has been instrumental in refining our work. We have diligently endeavored to address each of your recommendations to the best of our ability, striving to enhance the clarity, rigor, and overall quality of the paper.

Your expertise and guidance have been immensely valuable in shaping the manuscript into a more robust and coherent piece of research. We appreciate the time and effort you dedicated to the evaluation of our work.

Review 04:

1) Please improve figures quality. OK

Answer: Dear Reviewer,

I want to express my gratitude for all the considerations made regarding our work. I am confident that by following your suggestions, our work will reach a level of excellence that it would not have without them. The figures have been enhanced in quality following the guidelines of PLoS ONE. The graphs were extracted from MATLAB software, exported with the best available resolution, and verified for dimensions and quality using the PACE platform (PLoS ONE).

2) Please add proper flowchart. OK

Answer: Dear Reviewer,

I would like to thank you once again for this valuable contribution to improving our work. The flowchart has been appropriately added at the end of the paper in Appendix A, and it is also included in the supplementary materials, as explained at the end of the Materials and Methods section.

3) Please add block diagram. OK

Answer: Dear Reviewer,

The block diagram has been inserted in place of Figure 4. The block diagram highlights the circuit designed in Simulink software for determining the gains Kp, Ki, and Kd of the PID controller. Additionally, it shows the presence of the ISE and ISTE errors, which are calculated and sent to the script to compose the objective function.

4) Please justify gap and its addressing by your paper in literature. OK

Answer: Dear Reviewer,

Thank you for your insightful questions regarding our article. Your inquiries are elevating the quality of our study. We have added a paragraph at the end of the introduction to justify the gap in the literature that our study addresses. We focus on the lack of investigations into roll-off displacement with precise automatic adjustment control using bioinspired computational techniques such as PSO. Additionally, it aims to enable deployment of this control on hardware for real-time execution. In future work, we plan to integrate a dynamic actuation controller. This controller will receive information on tissue impedance variation and simultaneously adjust the PID controller to displace the roll-off, thus providing a more efficient ablation procedure.

Review 05:

It is a detailed theoretical and experimental study and the mostly the reviewers' comments are responded to; however, still, the following comments should be addressed:

1) Careful attention should be given to correcting writing mistakes. OK

Answer: Dear Reviewer,

Thank you for your feedback. We have carefully reviewed the paper and made the necessary corrections to address writing mistakes. Your input has been invaluable in improving the quality of our manuscript.

2) Although the introduction and related work have been presented, the differences of the study should have been briefly highlighted in related work. OK

Answer: Dear Reviewer,

Thank you for your consideration and suggestion regarding the topic in question. A paragraph has been added to the "Related Work" section briefly describing the differences in the study, as requested.

3) I recommend referencing the following publication for PSO and PID techniques: https://doi.org/10.1007/978-3-031-26876-2_85 . Additionally, more recent references from recent years should be included. OK

Answer: Dear Reviewer,

We greatly appreciate your suggestion. The study https://doi.org/10.1007/978-3-031-26876-2_85 has been included in the "Related Works" section as requested. Additionally, throughout the text, new references have been added to strengthen and substantiate our study in the literature.

4) In Fig 4, the peak point of the voltage, the stabil voltage, and applied voltage have not been explained in detail. The Voltage stabil time and later peak voltage should be explined. OK

Answer: Dear Reviewer,

Thank you for your valuable feedback. Regarding Figure 4, which depicts the voltage characteristics during our experiments, we have revised the text to provide a more detailed explanation of the peak point of the voltage, the stabilized voltage, and the applied voltage. We have also included an explanation of the voltage stabilization time and the later peak voltage. These changes aim to enhance the clarity and completeness of our presentation of the experimental data.

5) The meaning of the map and numbers in the root locus should be explained more thoroughly. OK

Answer: Dear Reviewer,

Thank you for your feedback. We have revised the explanation of the root locus diagram to provide a more thorough understanding of its meaning and the significance of the numbers displayed.

6) Overall ,except Figures 1 and 2, there is lack of detail in explanation for all the figures. OK

Answer: Dear Reviewer,

Thank you for your feedback regarding the figure captions. We have added information to the captions and improved how the information was conveyed. Additionally, we have enhanced the quality of all figures and included a new Figure 4, which caused all other figures to be renumbered accordingly.

Review 06:

1) The references provided are all applicable but not sufficient because there are some parts in the paper that need to be cited appropriately. OK

Answer: Dear Reviewer,

Thank you for your feedback. We have carefully reviewed the paper and added several references to address the points that needed to be cited appropriately. We believe these additions have significantly improved the paper's completeness and relevance to the existing literature.

---

## [Editor Report · Decision Letter 2]

28 Feb 2024

Particle swarm optimization solution for roll-off control in radiofrequency ablation of liver tumors: Optimal search for PID controller tuning

PONE-D-23-20054R2

Dear Dr. Faria,

We’re pleased to inform you that your manuscript has been judged scientifically suitable for publication and will be formally accepted for publication once it meets all outstanding technical requirements.

Kind regards,

Salim Heddam

Academic Editor

PLOS ONE

Additional Editor Comments (optional):

Reviewer #1:All comments have been addressed

Reviewer #6:

The paper presents a novel approach for controlling tissue impedance during RFA 710 procedures based on porcine liver ex-vivo experiments. The methodology combines 711 three system identification techniques, yielding superior results compared to individual 712 methods.

The paper is well written and organized in all sections, the results were well presented, likewise, the results from the analyses were discussed elaborately in an informative manner and were convincing, as well as the conclusion. The research novelty is somewhat minimal but scholarly convincing, as the discussions and analyses are based on existing technology but with little modification.

I hereby recommend the acceptance of this manuscript in its current form.

REMARKS

1. The title of the manuscript is appropriate.

2. The manuscript is technically sound and of High scientific quality.

3. The manuscript is free from errors and the grammar is satisfactory.

4. The tables and figures are clear.

5. The subject matter is presented comprehensively.

6. The references provided are all applicable but not sufficient because there are some parts in the paper that need to be cited appropriately.
---

## [Editor Report · Acceptance letter]

13 Jun 2024

PONE-D-23-20054R2 

PLOS ONE

Dear Dr. Faria, 

I'm pleased to inform you that your manuscript has been deemed suitable for publication in PLOS ONE. Congratulations! Your manuscript is now being handed over to our production team.

Kind regards, 

on behalf of

Dr. Salim Heddam 

Academic Editor

PLOS ONE